# A Rational Synthesis of a Branched Decaarabinofuranoside Related to the Fragments of Mycobacterial Polysaccharides

**DOI:** 10.3390/molecules30153295

**Published:** 2025-08-06

**Authors:** Polina I. Abronina, Nelly N. Malysheva, Maxim Y. Karpenko, Dmitry S. Novikov, Alexander I. Zinin, N. G. Kolotyrkina, Leonid O. Kononov

**Affiliations:** N.D. Zelinsky Institute of Organic Chemistry, Russian Academy of Sciences, Leninsky Prosp., 47, Moscow 119991, Russia; nelly.malyschewa@gmail.com (N.N.M.); m.karpenko96@mail.ru (M.Y.K.); darthvoron@gmail.com (D.S.N.); zinin@ioc.ac.ru (A.I.Z.); nkolotyr@ioc.ac.ru (N.G.K.)

**Keywords:** glycosylation, d-arabinofuranose, Janus aglycone, silyl group, *Mycobacterium tuberculosis*

## Abstract

A rational synthesis of the branched decaarabinofuranoside with 4-(2-azidoethoxy)phenyl aglycone (a Janus aglycone) related to the non-reducing terminal fragments of the arabinogalactan and lipoarabinomannan from *Mycobacterium tuberculosis* was proposed. Since the most challenging step is the formation of a 1,2-*cis* glycosidic linkage, we have significantly simplified access to a library of oligoarabinofuranosides derived from *Mycobacterium tuberculosis* polysaccharides using a silylated Ara-β-(1→2)-Ara disaccharide as the glycosyl donor. The application of a Janus aglycone also allowed us to reduce the number of reaction steps in glycoside synthesis. The obtained arabinans can be useful to further prepare conjugates as antigens for creating tuberculosis screening assays.

## 1. Introduction

The main causative agent of tuberculosis is *Mycobacterium tuberculosis* [1,2], and mycobacterial infections have received significant attention due to the increasing number of cases worldwide [3]. The components of the mycobacterial cell wall include arabinogalactan (AG) and lipoarabinomannan (LAM), which have a common motif consisting of α-(1→5), α-(1→3), and β-(1→2) glycosidic bonds. Due to the critical roles of AG and LAM in infectivity and pathogenicity of *Mycobacterium tuberculosis*, their structural fragments are currently attracting unprecedented attention for the study of pathways of biosynthesis of these glycopolymers [4], the development of diagnostics [5,6] and vaccines [7,8,9], as well as new effective antimicrobial drugs. It was found that neoglycoconjugates derived from synthetic fragments of LAM with bovine serum albumin (BSA), as well as neoglycoconjugates based on the non-reducing terminal branched hexasaccharide fragment (Ara_6_) with mycobacterial recombinant proteins, are important for the diagnosis of tuberculosis [6,10].

It should be noted that, despite many papers describing the preparation of oligoarabinofuranosides [11,12,13,14,15,16,17,18,19,20,21,22,23,24], the synthesis of the fragments of the mycobacterial cell wall remains a significant challenge. This is especially true for the preparation of oligoarabinofuranosides containing a large number of monosaccharide units (≥10), and the number of such studies remains limited.

It is known that 1,2-*trans*-α-arabinofuranosides can be easily synthesized using glycosyl donors with a participating acyl group at O-2. In contrast, the preparation of 1,2-*cis*-β-arabinofuranosides is more challenging, and 2-*O*-benzyl-containing glycosyl donors are often used for this purpose [13,16,18,20,21,22,24,25,26]. It should be noted that the removal of *O*-benzyl protective groups is incompatible with the azido group in the aglycone, which makes the synthesis less straightforward and requires additional steps [13].

We have recently showcased the benefits of glycosyl donors with O-2 protected by a triisopropylsilyl (TIPS) nonparticipating group (Figure 1, Block **A**, R = (*t*-Bu)_2_Si<) in the synthesis of the terminal oligoarabinoside fragments of *M. tuberculosis* such as penta- and hexaarabinofuranoside with α-(1→3)-, α-(1→5)- and β-(1→2)-linkages, containing 2-azidoethyl aglycone (Figure 2). Block **A** with a different type of remote protection was used by us in the synthesis of diarabinofuranosides Ara-β-(1→2)-Ara with 4-(2-azidoethoxy)phenyl (AEP) and 4-(3-azidopropoxy)phenyl (APP) aglycones. Moreover, the linear α-(1→5)-, β-(1→2)-linked tetrasaccharide AEP glycoside, as well as branched α-(1→3)-, α-(1→5)-, β-(1→2)-linked tetrasaccharide as an APP glycoside, were synthesized from Block **A**, (R = (*t*-Bu)_2_Si<) (Figure 2).

On the other hand, oligoarabinofuranosides with β-(1→2)-linked residues can be assembled using an Ara-β-(1→2)-Ara diarabinofuranoside glycosyl donor (Figure 1, Block **B**). In this case, the main difficulty is caused by the necessity to create a 1,2-*trans*-glycosidic linkage in the absence of a 2-*O*-acyl participating group. We have already demonstrated the successful stereoselective introduction of the silylated Ara-β-(1→2)-Ara disaccharide moiety (Figure 1, Block **B**), leading to more complex arabinans, including the formation of tetra [28] and branched hexaarabinofuranoside [30] **3** bearing APP aglycone and linear hexaarabinofuranoside with AEP aglycone [29] (Figure 2, Figure 1).

4-(ω-Chloroalkoxy)phenyl aglycones, due to their dual function, may be called Janus aglycones in analogy to the known nanoparticles [31]. 4-(ω-Chloroalkoxy)phenyl aglycones can be converted to 4-(ω-azidoalkoxy)phenyl arabinofuranosides [27]. The azido group in aglycone can be used for further preparation of corresponding glycosides with an amino group in aglycone [10] or it can be modified by using click-chemistry methods [32]. Moreover, both 4-(ω-chloroalkoxy)phenyl and 4-(ω-azidoalkoxy)phenyl glycosides are precursors of various glycosyl donors useful for the synthesis of more complex oligoarabinofuranosides [33].

It should be noted that the synthesis of oligosaccharides demanded selective multi-step protection and deprotection of numerous hydroxy groups and control of regio- and stereoselectivity. In this regard, developing strategies aimed at reducing the number of reaction steps still remains an important task. Since the most problematic glycosylation step is the creation of a 1,2-*cis*-glycosidic linkage, we significantly simplified the access to the library of oligoarabinofuranosides using Ara-β-(1→2)-Ara disaccharide as a glycosyl donor. Moreover, the application of 4-(ω-chloroalkoxy)phenyl aglycones also allows us to significantly reduce the number of reaction steps in glycoside synthesis.

Earlier, we synthesized branched hexaarabinofuranoside **1** with α-(1→3)-, α-(1→5)-, β-(1→2)-linkages and successfully converted it into deprotected AEP glycoside **3** for further conjugation [30] (Figure 1, Figure 1). However, the synthesis of hexaarabinofuranoside **1** by glycosylation of benzoylated diol **7** with diarabinofuranoside glycosyl donor **6** under NIS/Et_3_SiOTf promotion was complicated by the formation of Et_3_Si-substituted products of monoglycosylation of the glycosyl acceptor at both the primary and the secondary position.

In the current work, we aimed to test an alternative promotion system in the synthesis of branched hexaarabinofuranoside **2** containing the homologous 4-(2-chloroethoxy)phenyl (CEP) aglycone. Additionally, we focused on converting hexaarabinofuranoside **2** to a glycosyl donor (**2**→**4**) with subsequent preparation of deprotected decaarabinofuranoside **5** bearing AEP aglycone (Figure 1).

We hope that the application of our rational strategy based on the use of silylated Ara-β-(1→2)-Ara disaccharide **6** (Block **B**) will allow us easy access to the library of oligoarabinofuranosides. These oligosaccharides can then be used to create conjugates that act as antigens for tuberculosis screening assays.

## 2. Results

### 2.1. Synthesis of Branched α-(1→5)-, α-(1→3)-, β-(1→2)-Linked Hexaarabinofuranoside **2** with CEP Aglycone

#### Alternative Synthesis of the Known Silylated Ara-β-(1→2)-Ara p-Tolyl Thioglycoside **6** [28]

For the synthesis of hexaarabinofuranoside **2** with a 4-(2-chloroethoxy)phenyl (CEP) aglycone, similar to the preparation of hexaarabinofuranoside [30] **1** with a homologous 4-(3-chloropropoxy)phenyl (CPP) aglycone, we planned to use silylated Ara-β-(1→2)-Ara *p*-tolyl thioglycoside **6** with five TIPS groups [28]. In this study, we aim to investigate the conversion of silylated disaccharides **10** and **13**, which contain acetyl and CPP groups at the anomeric position, to Ara-β-(1→2)-Ara *p*-tolyl thioglycoside **6** (Figure 2). For this purpose, the previously obtained hemiacetal **9** [28] with five TIPS groups was acetylated with Ac_2_O/pyridine (Py), resulting in the single α-anomer of acetate **10**. Next, thiolysis of **10** was carried out with *p*-TolSH/BF_3_·Et_2_O at 40 °C for 40 min, followed by silylation of the resulting desilylation products. After silica gel chromatography, disaccharide **6** was isolated in 63% yield.

Alternatively, known disaccharide **11** with CPP aglycone was desilylated with TBAF in THF. The α-linked CPP-disaccharide **12**, isolated in 81% yield, contained 3% of substitution product with 4-(3-fluoropropoxy)phenyl aglycone. Moreover, silica gel chromatography failed to remove Bu_4_N^+^ (15% according to NMR). The resulting crude pentaol **12** was silylated by TIPSOTf in 2,4,6-collidine to give α-linked CPP-disaccharide **13**, also containing the product of substitution of a chlorine atom in the CPP aglycone with fluorine (3%). At the next stage, thiolysis of **13** with TolSH/BF_3_·Et_2_O was performed at −5 °C→0 °C for 2.5 h to give the target α-linked thioglycoside **6** isolated in a 50% yield. To the best of our knowledge, the products of the cleavage of the inter-saccharide glycosidic bond were not observed during thiolysis of **10** and **13**. In contrast, the use of the acetylated disaccharide Ara-β-(1→2)-Ara with a 4-(ω-chloroalkoxy)phenyl aglycone [28] resulted in significant amounts of products arising from cleavage of this bond.

### 2.2. Synthesis of Branched Hexaarabinofuranoside 2 with the Use of Silylated Ara-β-(1→2)-Ara p-Tolyl Thioglycoside **6**

At the next step, we performed the glycosylation of the diol of disaccharide with CEP aglycone **8** [28] with the Ara-β-(1→2)-Ara glycosyl donor **6** under AgOTf/NIS promotion. However, instead of the desired hexaarabinofuranoside **2**, the formation of α-(1→5)-linked tetraarabinofuranoside **14** was observed.

The anomeric signals of tetraarabinofuranoside **14** were found at δ_H_ 5.06 (s, 1H, H-1^III^), 5.20 (d, 1H, *J* 2.6 Hz, H-1^IV^), 5.37 (s, 1H, H-1^II^), and 5.83 (s, 1H, H-1^I^) in the ^1^H NMR spectrum, which correlated with signals at δ_C_ 104.8 (C-1^I^), 105.0 (C-1^IV^), 105.4 (C-1^II^), and 106.3 (C-1^III^) in the ^13^C NMR spectrum. An additional confirmation of the structure of tetraarabinofuranoside **14** and the formation of a new glycosidic linkage at the primary position followed from the fact that the ^1^H–^13^C HMBC spectrum showed a correlation between the signals of H-1^II^ at δ_H_ 5.37 ppm (s, 1H) and C-5^I^ at δ_C_ 66.0 ppm, a correlation between the signals of H-1^III^ at δ_H_ 5.06 ppm (s, 1H) and C-5^II^ at δ_C_ 64.4 ppm, and a correlation between the signals of H^IV^ at δ_H_ 5.20 (d, 1H, *J* 2.6 Hz) and C-2^III^ at δ_C_ 90.8 ppm.

Then, we introduced the second disaccharide Ara-β-(1→2)-Ara fragment at C-3^II^ of disaccharide **8** under TfOH/NIS promotion to form the key hexaarabinofuranoside **2** in 90% yield.

Moreover, we successfully obtained hexaarabinofuranoside **2** in a single step by bis-glycosylation of the diol **8** with *p*-tolyl thioglycoside Ara-β-(1→2)-Ara **6** under TfOH/NIS promotion in very high yield (97%) (Figure 3).

Comparison of the ^1^H and ^13^C NMR spectra for hexaarabinofuranosides with CPP and CEP aglycones (**1** and **2**, respectively), as expected, revealed many common features.

The signals of ten TIPS groups in the ^29^Si NMR spectrum for hexaarabinofuranosides **1**, **2** were observed in a similar range from δ_Si_ 12.3 to 13.9, and the signals of the corresponding carbon atoms of the isopropyl groups ((CH_3_)_2_CH)_3_Si) in the ^13^C NMR spectrum were observed in the range from δ_C_ 11.9 to 18.2. The signals of the isopropyl carbon atoms correlated with the ^1^H NMR proton signals observed at δ_H_ 0.86–1.15 (m, 210H, 10 × ((CH_3_)_2_CH)_3_Si).

The signals of the α-anomeric protons of the monosaccharide residues of hexaarabinofuranosides **2**, **1** are found at δ_H_ 5.10 (s, 1H, H-1^III^), 5.23 (s, 1H, H-1^V^), 5.27 (s, 1H, H-1^II^), 5.79 (s, 1H, H-1^I^), and 5.11 (s, 1H, H-1^III^), 5.23 (s, 1H, H-1^V^), 5.27 (s, 1H, H-1^II^), 5.80 (s, 1H, H-1^I^), respectively. While the following signals for the two β-anomeric protons of the monosaccharide residues of hexaarabinofuranosides **2**, **1** are present at δ_H_ 5.17 (d, 1H, *J* 2.5 Hz, H-1^VI^), 5.25 (d, 1H, *J* 2.8 Hz, H-1^IV^), and 5.17 (d, 1H, *J* 2.7 Hz, H-1^VI^), 5.25 (d, 1H, *J* 2.8 Hz, H-1^IV^), respectively.

The signals of the anomeric carbon atoms for all monosaccharide residues for hexaarabinofuranosides **2, 1** in the ^13^C NMR spectra appeared at δ_C_ 104.7 (C-1^IV^), 104.8 (C-1^I^), 105.4 (C-1^VI^), 106.0 (C-1^II^), 106.3 (C-1^V^), 106.9 (C-1^III^), and 104.69 (C-1^IV^), 104.71 (C-1^I^), 105.4 (C-1^VI^), 105.0 (C-1^II^), 106.3 (C-1^V^), 106.8 (C-1^III^), respectively. It is noteworthy that the low-field positions of the signals of C-1^IV^ and C-1^VI^ are untypical for β-arabinofuranosides, presumably due to the presence of bulky TIPS groups.

Moreover, similar correlations, confirming the formation of the glycosidic bonds between the corresponding monosaccharide residues in hexaarabinofuranosides **2**, **1** were observed in a ^1^H–^13^C HMBC spectrum: the correlation of the H-1^II^ proton signal at δ_H_ 5.27 (s, 1H) with the C-5^I^ carbon atom signal at δ_C_ 65.3; the correlation of the H-1^III^ proton signal at δ_H_ 5.1 (s, 1H) with the C-5^II^ carbon atom signal at δ_C_ 65.14; the correlation of the H-1^IV^ proton signal at δ_H_ 5.25 (d, 1H, *J* 2.8 Hz) with the C-2^III^ carbon atom signal at δ_C_ 90.5; and the correlation of the H-1^VI^ proton signal at δ_H_ 5.17 (d, 1H, *J* 2.5 Hz) with the C-2^V^ carbon atom signal at δ_C_ 91.3.

The H-2^III^ signal at δ_H_ 4.14 (d, 1H, *J* 0.8 Hz) correlates with the C-1^IV^ signal at δ_C_ 104.7; the H-2^III^ signal at 4.14 (d, 1H, *J* 1.1 Hz, H-2^III^) correlates with the C-1^IV^ signal at δ_C_ 104.7 for hexaarabinofuranosides **2** and **1**, respectively. Correlation of the H-2^V^ signal at 4.08 (d, 1H, *J* 1.2 Hz) with the C-1^VI^ signal at δ_C_ 105.4 and correlation of the H-2^V^ signal at 4.09 (d, 1H, *J* 1.3 Hz, H-2^V^) with the C-1^VI^ signal at δ_C_ 105.4 were also found for hexaarabinofuranosides **2** and **1**, respectively.

### 2.3. Synthesis of Branched α-(1→5)-, α-(1→3)-, β-(1→2)-Linked Decaarabinofuranoside **5** with 4-(2-Azidoethoxy)phenyl Aglycone

#### 2.3.1. Conversion of the Branched α-(1→5)-, α-(1→3)-, β-(1→2)-Linked Hexaarabinofuranoside **2** to Glycosyl Donor **4**

In the obtained hexaarabinofuranoside **2**, CEP aglycone was cleaved under oxidative conditions ((NH_4_)_2_Ce(NO_3_)_6_ in aqueous MeCN—CH_2_Cl_2_). After purification by silica gel column chromatography, hemiacetal **15** (70%) as a mixture of anomers (α:β = 1:0.45 according to NMR data), contaminated with the product **16** of migration of the benzoyl group from O-2 to O-1, was obtained (Figure 4). The ratio of the compounds in the mixture (**15**α:**15**β:**16** = 1:0.45:0.28 according to NMR data) was determined by integration of the signals of corresponding anomeric protons of the residues **I** in the ^1^H NMR spectrum: 5.62 (s, 1H, H-1^I^α) for **15α**, 5.72 (t, 1H, *J* 5.1 Hz, H-1^I^β) for **15β**, and 6.60 (d, 1H, *J* 4.5 Hz, H-1^I^) for **16**. The formation of the β-linked product **16** of the migration of the benzoyl group followed from the low-field position of the anomeric proton δ_H_ 6.60 (d, 1H, *J* 4.5 Hz, H-1^I^) in the ^1^H NMR spectrum, which correlated with the signal of anomeric carbon atom of monosaccharide residue δ_C_ 97.04 (C-1^I^) in the ^13^C NMR spectrum.

The mixture of hemiacetal **15** and β-linked benzoate **16** was treated with CF_3_C(NPh)Cl in the presence of Cs_2_CO_3_ (Figure 5), then subjected to gel chromatography on Bio-Beads S-X1 in toluene. *N*-Phenyltrifluoroacetimidate **4** was obtained as a mixture of anomers (α:β ~ 1:1.3 according to NMR data) in high yield (96%). It should be noted that the characteristic signal of β-linked benzoate **16** in the low-field position δ_H_ 6.60 (d, 1H, *J* 4.5 Hz, H-1^I^ minor) was absent in the ^1^H NMR spectrum. On the contrary, characteristic signals related to the *N*-phenyltrifluoroacetimidoyl group were present. The ratio of α and β isomers of *N*-phenyltrifluoroacetimidate **4** was determined by integrating the following signals in the ^1^H NMR spectrum: 6.51–6.57 (m, 2H, PhN (H-2, H-6)β) and 6.85–6.91 (m, 2H, PhN (H-2, H-6)α). Probably, a reverse migration of the benzoyl group from O-1 to O-2 for benzoate **16** took place under the basic conditions used for the imidate formation.

#### 2.3.2. Synthesis of the Protected Decaarabinofuranoside **18** with CEP Aglycone

The glycosylating ability of the obtained *N*-phenyltrifluoroacetimidate of hexaarabinofuranoside **4** was tested in the synthesis of branched α-(1→5)-, α-(1→3)-, β-(1→2)-linked decaarabinofuranoside **18** with cleavable CEP aglycone. To this end, benzoylated tetraarabinofuranoside α-(1→5)-linked glycosyl acceptor **17** [33] was glycosylated by *N*-phenyltrifluoroacetimidate **4** promoted by TfOH (Figure 5). After gel chromatography on Bio-Beads S-X1 in toluene, decaarabinofuranoside **18** was isolated in 72% yield (51% over 3 steps starting from **2**). It should be noted that the use of glycosyl donor **4** generated from **2** and having a 2-*O*-acyl participating group ensures the creation of 1,2-*trans*-(α) glycosidic linkage. The signals of ten TIPS groups were observed for decaarabinofuranoside in the ^29^Si NMR spectrum. It is important to note that no desilylation products were found despite the fact that TfOH was used as the promotor.

The anomeric configuration of decaarabinofuranoside **18** was confirmed by NMR spectroscopy. For α-linked Ara residues, the following signals of the anomeric protons were observed: δ_H_ 5.10 (s, 1H, H-1^VII^), 5.23 (s, 1H, H-1^IX^), 5.29 (s, 1H, H-1^VI^), 5.37 (s, 2H, 2 × H-1), 5.38–5.41 (m, 3H, H-2^VI^, 2 × H-1), 5.82 (s, 1H, H-1^I^). The following signals for two β-anomeric protons of the monosaccharide residues of decaarabinofuranoside **18** are present: δ_H_ 5.15 (d, 1H, *J* 2.6 Hz, H-1^X^) and δ_H_ 5.24 (d, 1H, *J* 2.8 Hz, H-1^VIII^). The signals of the anomeric carbon atoms for all monosaccharide residues for decaarabinofuranoside **18** as in case of branched hexaarabinofuranosides **1** [30], **2** and α-(1→5)-, β-(1→2)-linked tetraarabinofuranoside [28] resonated in a low-field region at δ_C_ 104.8 (C-1^VIII^), 104.9 (C-1^I^), 105.5 (C-1^X^), 105.85 (C-1^VI^), 105.89 (C-1), 106.0 (3C, C-1^II^, C-1^III^, C-1^IV^), 106.2 (C-1^IX^), and 106.9 (C-1^VII^) in the ^13^C NMR spectra. Moreover, correlations confirming the presence of glycosidic bonds between the corresponding monosaccharide residues of decaarabinofuranoside **18** were found in its ^1^H–^13^C HMBC spectrum: the H-1^VI^ signal at δ_H_ 5.29 (s, 1H) correlates with the C-5^V^ signal at δ_C_ 65.3; the H-1^VII^ signal at δ_H_ 5.10 (s, 1H) correlates with the C-5^VI^ signal at δ_C_ 64.8; the H-1^VIII^ signal at δ_H_ 5.24 (d, 1H, *J* 2.8 Hz, H-1^VIII^) correlates with the C-2^VII^ signal at δ_C_ 90.6; the H-1^IX^ signal at δ_H_ 5.23 (s, 1H, H-1^IX^) correlates with the C-3^VI^ signal at δ_C_ 81.0 (C-3^VI^); and the H-1^X^ signal at δ_H_ 5.15 (d, 1H, *J* 2.6 Hz, H-1^X^) correlates with the C-2^IX^ signal at δ_C_ 91.4 (C-2^IX^).

#### 2.3.3. Synthesis of the Branched α-(1→5)-, α-(1→3)-, β-(1→2)-Linked Decaarabinofuranoside **5** with AEP Aglycone

The chlorine atom in the aglycone of the resulting decaarabinofuranoside **18** was replaced with an azido group (NaN_3_, DMF, 18-crown-6) to give the corresponding AEP glycoside **19** in 84% yield (Figure 5). Then all TIPS groups in azide **19** were removed by treatment with TBAF in THF in the presence of AcOH at 40 °C to give partially protected decaarabinofuranoside **20** isolated in a mixture with *n*-Bu_4_N^+^ salts after silica gel chromatography, which was treated with MeONa in MeOH with subsequent acetylation (Figure 6). After purification by silica gel chromatography, acetylated decasaccharide **21** (14% over 3 steps) was obtained. The overall low deprotection yield can be attributed to the difficulty of the removal of the numerous (ten) triisopropylsilyl groups, followed by purification from by-products.

The anomeric configuration of acetylated decasaccharide **21** was verified by NMR spectroscopy. As expected, the signals of the two β-anomeric protons were found at δ_H_ 5.39 (d, 1H, *J* 4.7 Hz, H-1^X^) and δ_H_ 5.44 (d, 1H, *J* 4.7 Hz, H-1^VIII^), which correlated with the signals of the anomeric carbon atoms in the characteristic regions at δ_C_ 99.29 (C-1^X^) and 99.50 (C-1^VIII^), respectively. The obtained decasaccharide **21** was deacetylated by treatment with MeONa in MeOH to give the target deprotected decasaccharide **5** in 85% yield.

The presence of two β-anomeric Ara^VIII^ and Ara^X^ residues of the target deprotected decasaccharide **5** was confirmed by the signals of the C-1^VIII^ and C-1^X^ carbon atoms at δ_C_ 102.5, 102.6 (C-1^VIII^, C-1^X^). In the HSQC spectrum, the C-1^VIII^ and C-1^X^ carbon atom signals correlated with the proton signals at δ_H_ 5.02–5.06 (m, 2H, H-1^VIII^, H-1^X^). The signals of the eight α-linked arabinofuranose moieties were observed in the characteristic low-field regions at δ_C_ 107.1 (C-1^IX^), 107.4 (C-1^VII^), 108.7 (C-1^I^), 109.58, 109.63, 109.70 (2C), and 109.74 (C-1^II-VI^). The upfield position of C-2^VIII^ and C-2^X^ at δ_C_ 79.19 additionally confirms the β-configuration of the residues **VIII** and **X**. On the contrary, the low-field position of C-2^VII^, C-2^IX^ at δ_C_ 89.3 (C-2^VII^), 89.6 (C-2^IX^), and C-3^VI^ at δ_C_ 84.6, respectively, indicated that those residues are glycosylated.

## 3. Discussion

We suggested an alternative synthesis of diarabinofuranoside **6** and found that during the thiolysis of silylated disaccharides Ara-β-(1→2)-Ara (**10** and **13**) containing acetyl or CPP groups at the anomeric position, no products of cleavage of the inter-saccharide glycosidic bond were observed.

It is important to note that the previously proposed synthesis of **6** included the reaction of acetylated Ara-β-(1→2)-Ara bearing 4-(ω-chloroalkoxy)phenyl aglycone with TolSH/BF_3_·Et_2_O, which required a higher temperature and a longer time (20 °C for 16 h, then 60 °C for 2 h). Moreover, thiolysis of acetylated Ara-β-(1→2)-Ara was complicated by the formation of monosaccharides [28]. We can conclude that the protective groups in Ara-β-(1→2)-Ara have a significant effect on the course of the thiolysis. Thus, the presence of electron-donating TIPS groups is favorable for the reaction.

We tested various promotor systems, such as NIS/TfOH, NIS/TESOTf [30], and NIS/AgOTf to activate silylated Ara-β-(1→2)-Ara *p*-tolyl thioglycoside **6** in the synthesis of branched hexasaccharide **2**. No glycosylation of the secondary 3′-OH of diarabinofuranoside glycosyl acceptor **14** was observed under mild activation conditions (NIS/AgOTf), and only α-(1→5)-linked tetraarabinofuranoside **14** was obtained. On the contrary, the use of a stronger activation system (NIS/TfOH) made possible stereospecific glycosylation of *both* primary (5′-OH) and secondary (3′-OH) hydroxy groups and afforded the desired hexaarabinofuranoside **2** in exceptionally high yield. We can conclude that the use of (NIS/TfOH) is more preferable for bis-glycosylation.

It should be noted that, like in the case of hexaarabinofuranoside **1** with the CPP aglycone, an exclusively α-configuration of the glycosidic linkage was observed in hexaarabinofuranoside **2** bearing the CEP aglycone. On the contrary, as we showed earlier, in the case of the use of Ara-β-(1→2)-Ara disaccharide glycosyl donor, which contains only *O*-benzoyl substituents, the absence of stereocontrol (α:β = 1:2) was observed [28].

TfOH also effectively facilitated the reaction between *N*-phenyltrifluoroacetimidate of hexaarabinofuranoside **4** and α-(1→5)-linked benzoylated tetrasaccharide glycosyl acceptor **17**, resulting in the formation of decaarabinofuranoside **18**. As we mentioned above, the nature of the protective group in the glycosyl donor is very essential in the outcome of glycosylation. Recently, we observed an unusual oligomerization of arabinofuranosides promoted by TfOH during the glycosylation of the primary hydroxyl group of the same α-(1→5)-linked tetraarabinofuranoside **17**, which bears a 4-(2-chloroethoxy)phenyl aglycone by benzoylated *N*-phenyltrifluoroacetimidate [33]. To explain these unusual results, we suggested that silylated disaccharide Ara-β-(1→2)-Ara glycosyl donor **6** has superior reactivity compared to “disarmed” benzoylated glycosyl donors. Due to the higher reactivity of the former, side reactions are suppressed, and we did not observe the formation of the oligomerization products.

As the most challenging step in glycosylation is creating a 1,2-*cis* glycosidic linkage, we have significantly simplified access to a library of oligoarabinofuranosides from *Mycobacterium tuberculosis* using silylated Ara-β-(1→2)-Ara disaccharide. In addition, the use of a 4-(2-chloroethoxy)phenyl aglycone allowed us to reduce the number of reaction steps in glycosidic synthesis.

## 4. Materials and Methods

**General methods.** All reactions sensitive to air and/or moisture were carried out under an argon atmosphere. The reactions were performed with the use of commercial reagents (Aldrich (St. Louis, MO, USA), Fluka (Waltham, MA, USA), Acros Organics (Geel, Belgium)). Anhydrous solvents were purified and dried (where appropriate) according to standard procedures [34]. Dichloromethane was distilled over P_2_O_5_ and then over CaH_2_ and stored over 4 Å molecular sieves (MS 4 Å). Powdered MS 4 Å and 3 Å molecular sieves (MS 4 Å and MS 3 Å, respectively) (Fluka, Seelze, Germany) were activated before glycosylation reactions by heating at 220 °C in high vacuum (0.2 mbar) for 6 h. Column chromatography was performed on silica gel 60 (40–63 μm, Merck, Darmstadt, Germany) using a Büchi C-815 Flash chromatograph (Büchi Labotechnic AG, Flawil, Switzerland). Thin-layer chromatography was carried out on plates with silica gel 60 on aluminum foil (Merck). Spots of compounds were visualized under UV light (254 nm) and by heating the plates (at ca. 150 °C) after immersion in a 1:10 (*v*/*v*) mixture of 85% aqueous H_3_PO_4_ and 95% EtOH. Gel permeation chromatography was performed on a 400 × 20 mm column packed with Bio-Beads S-X3 (200–400 mesh, Bio-Rad, Hercules, CA, USA) or on a 450 × 30 mm column packed with Bio-Beads S-X1 (200–400 mesh). A procedure for “co-evaporation” with water (or toluene) involved (multiple) addition of water (or toluene) and evaporation of volatiles on a rotary evaporator. Amberlite MB-3 mixed-bed ion-exchange resin (Fluka) (1 mL) was washed with H_2_O (10 mL), 50% EtOH (5 mL), EtOH (5 mL), 50% EtOH (5 mL), and H_2_O (10 mL) before use. ^1^H, ^13^C, ^19^F, and ^29^Si NMR spectra were recorded on a Bruker AVANCE NEO 300 spectrometer (300.23, 75.50, 282.47, and 59.65 MHz for ^1^H, ^13^C, ^19^F, and ^29^Si, respectively) or on a Bruker AVANCE 600 spectrometer (Billerica, MA, USA, 600.13 and 150.92 MHz for ^1^H and ^13^C, respectively). The ^1^H NMR chemical shifts are referred to the residual signal of CHCl_3_ (δ_H_ 7.27 ppm), CHD_2_OD (δ_H_ 3.31 ppm) for solutions in CD_3_OD, and HDO (δ_H_ 4.79 ppm) for solutions in D_2_O. The ^13^C NMR shifts to the central line of the CDCl_3_ signal (δ_C_ 77.00 ppm), CD_3_OD signal (δ_C_ 49.00 ppm), or the signal of external 1,4-dioxane in D_2_O (δ_C_ 67.40 ppm). The ^19^F chemical shifts are given relative to the signal of external CFCl_3_ (δ_F_ 0.00 ppm). The ^29^Si chemical shifts are given relative to the signal of external Me_4_Si (δ_Si_ 0.00 ppm). Assignments of the signals in the NMR spectra were performed using ^1^H–^1^H and ^1^H–^13^C 2D-spectroscopy (COSY, HSQC, HMBC) and DEPT-135 experiments. Position of silyl groups was determined from ^1^H–^29^Si HMBC experiments. High-resolution mass spectra (HRMS, electrospray ionization (ESI)) were recorded in a positive ion mode on Bruker micrOTOF II or maXis mass spectrometers for 2 × 10^−5^ M solutions in MeCN. Optical rotations were measured using a JASCO P-2000 automatic digital polarimeter (Hachioji, Japan).


**4-(2-Chloroethoxy)phenyl 2,3,5-tris-*O*-(triisopropylsilyl)-β**
**-d**
**-arabinofuranosyl-(1→**
**2)-3,5-bis-*O*-(triisopropylsilyl)-α**
**-d-**
**arabinofuranosyl-(1→**
**3)-[2,3,5-tris-*O*-(triisopropylsilyl)-β**
**-d**
**-arabinofuranosyl-(1→**
**2)-3,5-bis-*O*-(triisopropylsilyl)-α**
**-d-**
**arabinofuranosyl-(1→**
**5)]-2-*O*-benzoyl-α**
**-d-**
**arabinofuranosyl-(1→**
**5)-2,3-di-*O*-benzoyl-α**
**-d-**
**arabinofuranoside (2)**


(1) A mixture of disaccharide thioglycoside **6** (37 mg, 0.31 mmol) and tetrasaccharide alcohol **14** (19 mg, 0.25 mmol) was dried in vacuo for 2 h, then anhydrous CH_2_Cl_2_ (2 mL) was added under argon. Freshly activated powdered MS 4 Å (200 mg) (100 mg per 1 mL of solvent) was added under argon to the resulting solution. The suspension was stirred under argon at ~22 °C for 1 h, cooled to −60 °C, then NIS (7 mg, 0.31 mmol) and TfOH (1 μL) were added. Then, the temperature was allowed to rise slowly until −30 °C during 0.5 h and was kept at −30 °C for 10 min. Then the reaction was quenched by the addition of Py (50 μL), diluted with CH_2_Cl_2_ (15 mL), and filtered through a Celite pad. The solids were washed with CH_2_Cl_2_ (5 × 10 mL), and the filtrate was washed with a mixture of satd aq Na_2_S_2_O_3_ (50 mL) and satd aq NaHCO_3_ (50 mL). The aqueous layer was extracted with CH_2_Cl_2_ (2 × 5 mL). The combined organic extracts were filtered through a cotton wool plug, concentrated and dried in vacuo, then dissolved in toluene (2 mL) and subjected to chromatography on Bio-Beads S-X1 in toluene to give α-linked hexasaccharide **2** (52 mg, 90%); *R*_f_ = 0.50 (light petroleum–EtOAc 10:1); ${\left[\alpha \right]}_{D}^{28}$ +31.0 (c 1.0 in CHCl_3_); ^1^H NMR (600 MHz, CDCl_3_): δ 0.95–1.15 (m, 210H, 10 × ((CH_3_)_2_CH)_3_Si), 3.63 (dd, 1H, *J* 9.5 Hz, *J* 4.5 Hz, H-5^IV^a or H-5^VI^a), 3.66 (dd, 1H, *J* 10.5 Hz, *J* 5.4 Hz, H-5^V^a), 3.71 (dd, 1H, *J* 9.5 Hz, *J* 4.5 Hz, H-5^IV^a or H-5^VI^a), 3.78 (t, 2H, *J* 6.0 Hz, CH_2_Cl), 3.73–3.99 (m, 11H, 8 × H-5, H-4^IV^, H-4^VI^, H-4^V^), 4.03–4.05 (m, 2H, H-2^IV^, H-2^VI^), 4.09 (d, 1H, *J* 1.3 Hz, H-2^V^), 4.11 (td, 1H, *J* 5.9 Hz, *J* 3.9 Hz, H-4^III^), 4.14 (d, 1H, *J* 1.1 Hz, H-2^III^), 4.15–4.19 (m, 1H, H-5^I^b), 4.17 (t, 2H, *J* 6.0 Hz, CH_2_O), 4.22 (dd, 1H, *J* 7.0 Hz, *J* 2.1 Hz, H-3^II^), 4.30 (d, 1H, *J* 0.9 Hz, H-3^VI^), 4.29–4.32 (m, 1H, H-4^II^), 4.33 (d, 1H, *J* 0.8 Hz, H-3^IV^), 4.34 (d, 1H, *J* 4.0 Hz, H-3^III^), 4.45 (dd, 1H, *J* 3.7 Hz, *J* 1.4 Hz, H-3^V^), 4.60 (q, 1H, *J* 4.2 Hz, H-4^I^), 5.11 (s, 1H, H-1^III^), 5.17 (d, 1H, *J* 2.7 Hz, H-1^VI^), 5.23 (s, 1H, H-1^V^), 5.25 (d, 1H, *J* 2.8 Hz, H-1^IV^), 5.27 (s, 1H, H-1^II^), 5.41 (d, 1H, *J* 2.1 Hz, H-2^II^), 5.65 (dd, 1H, *J* 4.5 Hz, *J* 2.0 Hz, H-3^I^), 5.71 (d, 1H, *J* 1.6 Hz, H-2^I^), 5.80 (s, 1H, H-1^I^), 6.82–6.87 (m, 2H, OC_6_H_4_O (H-3, H-5)), 7.03–7.09 (m, 2H, OC_6_H_4_O (H-2, H-6)), 7.38–7.44 (m, 2H, 2^II^-O-PhCO (H-3, H-5)), 7.44–7.62 (m, 7H, PhCO), 8.01–8.05 (m, 2H, 2^II^-O-PhCO (H-2, H-6)), 8.10–8.13 (m, 2H, PhCO (H-2, H-6)), 8.13–8.16 (m, 2H, PhCO (H-2, H-6)); ^13^C NMR (151 MHz, CDCl_3_): δ 11.93 (((CH_3_)_2_CH)_3_Si), 11.96 (((CH_3_)_2_CH)_3_Si), 11.97 (((CH_3_)_2_CH)_3_Si), 12.01 (((CH_3_)_2_CH)_3_Si), 12.18 (((CH_3_)_2_CH)_3_Si), 12.23 (((CH_3_)_2_CH)_3_Si), 12.24 (2 × ((CH_3_)_2_CH)_3_Si), 12.3 (((CH_3_)_2_CH)_3_Si), 12.4 (((CH_3_)_2_CH)_3_Si), 17.9 (2 × ((CH_3_)_2_CH)_3_Si), 17.95 (2 × ((CH_3_)_2_CH)_3_Si), 17.96 (2 × ((CH_3_)_2_CH)_3_Si), 18.02 (2 × ((CH_3_)_2_CH)_3_Si), 18.03 (((CH_3_)_2_CH)_3_Si), 18.05 (((CH_3_)_2_CH)_3_Si), 18.08 (((CH_3_)_2_CH)_3_Si), 18.09 (2 × ((CH_3_)_2_CH)_3_Si), 18.11 (((CH_3_)_2_CH)_3_Si), 18.12 (4 × ((CH_3_)_2_CH)_3_Si), 18.17 (((CH_3_)_2_CH)_3_Si), 18.19 (((CH_3_)_2_CH)_3_Si), 41.9 (CH_2_Cl), 63.7 (C-5^IV^, C-5^VI^), 63.8 (C-5^IV^, C-5^VI^), 64.1 (C-5^V^), 64.7 (C-5^III^), 65.1 (C-5^II^), 65.3 (C-5^I^), 68.7 (CH_2_O), 77.5 (C-3^I^), 77.6 (C-3^V^), 78.1 (C-2^IV^ or C-2^VI^), 78.2 (C-3^VI^), 78.3 (C-2^IV^ or C-2^VI^), 78.48 (C-3^III^, C-3^IV^), 78.51 (C-3^III^, C-3^IV^), 81.0 (C-3^II^), 81.9 (C-2^I^), 82.2 (C-4^II^), 82.9 (C-4^I^), 84.0 (C-2^II^), 85.68 (C-4^IV^, C-4^VI^), 85.74 (C-4^IV^, C-4^VI^), 87.6 (C-4^III^), 88.2 (C-4^V^), 90.5 (C-2^III^), 91.3 (C-2^V^), 104.69 (C-1^IV^), 104.71 (C-1^I^), 105.7 (C-1^VI^), 106.0 (C-1^II^), 106.3 (C-1^V^), 106.8 (C-1^III^), 115.8 (OC_6_H_4_O (C-3, C-5)), 118.3 (OC_6_H_4_O (C-2, C-6)), 128.2 (2^II^-*O*-PhCO (C-3, C-5)), 128.4 (PhCO (C-3, C-5)), 128.6 (PhCO (C-3, C-5)), 129.1 (PhCO (C-1)), 129.5 (PhCO (C-1)), 129.81 (PhCO (C-1)), 129.84 (2^II^-*O*-PhCO (C-2, C-6)), 129.95 (PhCO (C-2, C-6)), 130.03 (PhCO (C-2, C-6)), 132.8 (2^II^-*O*-PhCO (C-4)), 133.2 (PhCO (C-4)), 133.4 (PhCO (C-4)), 151.0 (OC_6_H_4_O (C-1)), 153.5 (OC_6_H_4_O (C-4)), 165.1 (2^II^-*O*-PhCO), 165.5 (PhCO), 165.6 (PhCO); ^29^Si INEPT NMR (60 MHz, CDCl_3_): δ 12.36, 12.93, 13.26, 13.48, 13.51, 13.53, 13.56, 13.79, 13.83, 13.87; HRMS (ESI): *m/z* [M + 2NH_4_]^2+^ Calcd for C_149_H_277_ClN_2_O_29_Si_10_^2+^ 2855.7294; Found: 2855.7251;

(2) A mixture of disaccharide thioglycoside **6** (127 mg, 0.11 mmol) and disaccharide **8** (22 mg, 0.03 mmol) was dried in vacuo for 2 h, then anhydrous CH_2_Cl_2_ (3 mL) was added under argon. Freshly activated powdered MS 4 Å (300 mg) (100 mg per 1 mL of solvent) was added under argon to the resulting solution. The suspension was stirred under argon at ~22 °C for 1 h, cooled to −60 °C, then NIS (7 mg, 0.31 mmol) and TfOH (1 μL) were added. Then, the temperature was allowed to rise slowly to −30 °C during 0.5 h and was kept at −30 °C for 10 min. Then the reaction was quenched by the addition of Py (50 μL), diluted with CH_2_Cl_2_ (15 mL), and filtered through a Celite pad. The solids were washed with CH_2_Cl_2_ (5 × 10 mL), and the filtrate was washed with a mixture of satd aq Na_2_S_2_O_3_ (50 mL) and satd aq NaHCO_3_ (50 mL). The aqueous layer was extracted with CH_2_Cl_2_ (2×5 mL). The combined organic extracts were filtered through a cotton wool plug, concentrated and dried in vacuo, then dissolved in toluene (2 mL) and subjected to chromatography on Bio-Beads S-X1 in toluene to give hexasaccharide **2** (80 mg, 97%).


**2,3,5-Tris-*O*-(triisopropylsilyl)-β**
**-d**
**-arabinofuranosyl-(1→**
**2)-3,5-bis-*O*-(triisopropylsilyl)-α**
**-d-**
**arabinofuranosyl-(1→**
**3)-[2,3,5-tris-*O*-(triisopropylsilyl)-β**
**-d**
**-arabinofuranosyl-(1→**
**2)-3,5-bis-*O*-(triisopropylsilyl)-α**
**-d-**
**arabinofuranosyl-(1→**
**5)]-2-*O*-benzoyl-α**
**-d-**
**arabinofuranosyl-(1→**
**5)-2,3-di-*O*-benzoyl-**
**
d-
**
**arabinofuranosyl *N*-phenyltrifluoroacetimidate (4)**


To a solution of hemiacetal of hexasaccharide **15** (31 mg, 0.012 mmol) in CH_2_Cl_2_ (1 mL), Cs_2_CO_3_ (30 mg, 0.093 mmol) and CF_3_C(NPh)Cl [35] (4 μL, 0.023 mmol) were added at 0 °C. The reaction mixture was stirred at 20 °C for 3h. The reaction mixture was diluted with CH_2_Cl_2_ (20 mL) and then filtered through the cotton wool plug. The filtrate concentrated under reduced pressure, and the residue was dried in vacuo and subjected to gel chromatography on Bio-Beads S-X1 in toluene to give imidate **4** as a mixture of anomers (α:β = ~1:1.3 according to NMR), (32 mg, 96%). *R*_f_ = 0.76 (light petroleum–EtOAc, 8.5:1.5); selected signals: ^1^H NMR (300 MHz, CDCl_3_): δ 0.78–1.35 (m, 483H), 3.61–4.01 (m, 38H), 4.01–4.19 (m, 16H), 4.19–4.39 (m, 13H), 4.43–4.48 (m, 2H), 4.51–4.59 (m, 1H), 4.61–4.71 (m, 1H), 5.08 (s, 1H), 5.10 (s, 1H), 5.16–5.28 (m, 11H), 5.36 (d, 1H, *J* 1.9 Hz), 5.62 (d, 1H, *J* 3.7 Hz), 5.73 (s, 1H), 5.84–5.97 (m, 2H), 6.51–6.57 (m, 2H, PhN (H-2, H-6)β), 6.79 (s, 1H, H-1), 6.85–6.91 (m, 2H, PhN (H-2, H-6)α), 6.98–7.10 (m, 2H, PhN (H-4)), 7.12–7.20 (m, 2H, PhN (H-3, H-5)), 7.22–7.31 (m, 5H, PhCO (H-3, H-5)), 7.36–7.63 (m, 25H, PhCO (H-3, H-5); PhCO (H-4), 7.97–8.17 (m, 16H, PhCO (H-2, H-6)); ^13^C NMR (76 MHz, CDCl_3_): δ 12.0 (((CH_3_)_2_CH)_3_Si), 12.2 (((CH_3_)_2_CH)_3_Si), 12.4 (((CH_3_)_2_CH)_3_Si), 18.0 (((CH_3_)_2_CH)_3_Si), 18.1 (((CH_3_)_2_CH)_3_Si), 63.7, 63.8, 64.1, 64.6, 64.7, 67.8, 78.3, 78.4, 83.7, 85.7, 87.6, 88.2, 90.8, 91.2, 97.3 (C-1), 105.4 (C-1), 105.89 (C-1), 106.91 (C-1), 119.3 (PhN (C-2, C-6)), 119.7 (PhN (C-2, C-6)), 124.0, 128.2, 128.4, 128.49, 128.53, 128.6, 129.85 (PhCO (C-2, C-6)), 129.94 (PhCO (C-2, C-6)), 130.0 (PhCO (C-2, C-6)), 132.9 (PhCO (C-4)), 133.4 (PhCO (C-4)), 133.6 (PhCO (C-4)), 165.1 (CO), 165.4 (CO); ^29^Si INEPT NMR (60 MHz, CDCl_3_): δ 12.39, 12.45, 12.84, 12.94, 13.26, 13.38, 13.47, 13.50, 13.53, 13.55, 13.74, 13.79, 13.83, 13.88, 13.91; ^19^F NMR (282 MHz, CDCl_3_): δ −65.76 (CF_3_); HRMS (ESI): *m*/*z* [M + NH_4_]^+^ Calcd for C_149_H_270_F_3_N_2_O_28_Si_10_^+^ 2872.7404; found: 2872.7378.


**4-(2-Azidoethoxy)phenyl β**
**-d**
**-arabinofuranosyl-(1→**
**2)-α**
**-d-**
**arabinofuranosyl-(1→**
**3)-[β**
**-d**
**-arabinofuranosyl-(1→**
**2)-α**
**-d-**
**arabinofuranosyl-(1→**
**5)]-α**
**-d-**
**arabinofuranosyl-(1→**
**5)-α**
**-d-**
**arabinofuranosyl-(1→**
**5)-α-d-arabinofuranosyl-(1→**
**5)-α-d-arabinofuranosyl-(1→**
**5)-α-d-arabinofuranosyl-(1→**
**5)-α-d-arabinofuranoside (5)**


Acetylated decasaccharide **21** (1.5 mg, 0.0006 mmol) was dissolved in anhydrous CH_2_Cl_2_ (0.2 mL) and anhydrous MeOH (0.8 mL), followed by the addition of 1 M methanolic MeONa (20 μL). The reaction mixture was kept at ~20 °C for 24 h. Then, the reaction mixture was neutralized with Dowex 50W X8 (H^+^) ion-exchange resin (the resin was washed with MeOH before addition) and then filtered. The resin was washed with MeOH (3 × 20 mL). The combined filtrate was concentrated under reduced pressure. The residue was co-evaporated with toluene (2 × 2 mL), dried in vacuo, dissolved in water (1 mL), and applied on a small column (ID 5 mm) packed with Amberlite MB-3 mixed-bed ion-exchange resin (1 mL), which was eluted with H_2_O (15 mL). The eluate was lyophilized and additionally purified by reversed phase chromatography on a Sep-Pak C18 cartridge (particle size: 55–105 μm, pore size: 125 Å, sorbent substrate: silica, sorbent weight: 360 mg), gradient: 0→100% MeCN in H_2_O) to give deprotected decasaccharide **5** (0.8 mg, 85%). *R*_f_ = 0.50 (MeCN–H_2_O 7.5:2.5); ${\left[\alpha \right]}_{D}^{28}$ +15.1 (c 0.2 in MeOH); ^1^H NMR (600 MHz, CD_3_OD): δ 3.54–3.58 (m, 2H, CH_2_N_3_), 3.61–3.88 (m, 21H, H-5^I-X^a, H-4^VIII, X^, H-5^I-V, VII-X^b), 3.89–3.95 (m, 5H, H-5^VI^b, H-3^II-V^), 3.99–4.06 (m, 12H, H-2^II-V, VIII, X^, H-3^I, VI-X^), 4.06–4.11 (m, 4H, H-4^II-V^), 4.12 (dd, 2H, *J* 5.4 Hz, *J* 4.4 Hz, CH_2_O), 4.13–4.15 (m, 3H, H-2^VII^, H-2^IX^, H-4^I^), 4.16 (dd, 1H, *J* 2.7 Hz, *J* 1.5 Hz, H-2^VI^), 4.17–4.21 (m, 1H, H-4^VI^), 4.21 (dd, 1H, *J* 4.0 Hz, *J* 1.8 Hz, H-2^I^), 4.94–4.98 (m, 5H, H-1^II-VI^), 5.02–5.06 (m, 2H, H-1^VIII^, H-1^X^), 5.09 (d, 1H, *J* 2.2 Hz, H-1^VII^), 5.17 (d, 1H, *J* 2.4 Hz, H-1^IX^), 5.42 (d, 1H, *J* 2.0 Hz, H-1^I^), 6.86–6.91 (m, 2H, OC_6_H_4_O (H-3, H-5)), 6.97–7.03 (m, 2H, OC_6_H_4_O (H-2, H-6)); ^13^C NMR (151 MHz, CD_3_OD): δ 51.4 (CH_2_N_3_), 62.45 (C-5^VII^, C-5^IX^), 62.53 (C-5^VII^, C-5^IX^), 64.38 (C-5^VIII^, C-5^X^), 64.41 (C-5^VIII^, C-5^X^), 67.9 (2C, C-5^I-VI^), 68.0 (C-5^I-VI^), 68.3 (3C, C-5^I-VI^), 69.0 (CH_2_O), 75.8 (C-3^VIII^, C-3^X^), 75.9 (C-3^VIII^, C-3^X^), 76.3 (C-3^VII^, C-3^IX^), 76.5 (C-3^VII^, C-3^IX^), 78.8 (3C, C-2^VIII^, C-2^X^, C-3^I-V^), 78.9 (C-2^VIII^, C-2^X^, C-3^I-V^), 79.2 (3C, C-2^VIII^, C-2^X^, C-3^I-V^), 81.6 (C-2^VI)^, 83.03 (C-4^VI^), 83.3 (C-2^II-V^), 83.8 (C-2^I^), 83.95 (C-4^I-V^, C-4^VII-X^), 84.03 (C-4^I-V^, C-4^VII-X^), 84.2 (4C, C-4^I-V^, C-4^VII-X^), 84.3 (C-4^I-V^, C-4^VII-X^), 84.4 (2C, C-4^I-V^, C-4^VII-X^), 84.6 (C-3^VI^), 89.3 (C-2^VII^), 89.6 (C-2^IX^), 102.5 (C-1^VIII^, C-1^X^), 102.6 (C-1^VIII^, C-1^X^), 107.1 (C-1^IX^), 107.4 (C-1^VII^), 108.7 (C-1^I^), 109.58 (C-1^II-VI^), 109.63 (C-1^II-VI^), 109.70 (2C, C-1^II-VI^), 109.74 (C-1^II-VI^), 116.6 (OC_6_H_4_O (C-3, C-5)), 119.3 (OC_6_H_4_O (C-2, C-6)), 152.7 (OC_6_H_4_O (C-1)), 155.2 (OC_6_H_4_O (C-4)); HRMS (ESI): *m/z* [M + NH_4_]^+^ Calcd for C_58_H_93_N_4_O_42_^+^ 1517.5259; found: 1517.5244.


**
*p*
**
**-Tolyl 2,3,5-tris-*O*-(triisopropylsilyl)-β**
**-d**
**-arabinofuranosyl-(1→**
**2)-1-thio-3,5-bis-*O*-(triisopropylsilyl)-α**
**-d-**
**arabinofuranoside (6)**


(1) To a solution of acetate **10** (41 mg, 0.037 mmol) in ClCH_2_CH_2_Cl (1 mL), *p*-TolSH (9 mg, 0.074 mmol) and BF_3_·Et_2_O (2 μL, 0.02 mmol) were added at 20 °C. The reaction mixture was stirred at 40 °C for 1 h. Then, the reaction mixture was diluted with CH_2_Cl_2_ (50 mL) and washed with H_2_O (50 mL). The organic phase was filtered through a cotton wool plug, concentrated, and dried in vacuo. The residue was dissolved in 2,4,6-collidine (1 mL), then *i*-Pr_3_SiOTf (450 mL, mmol) was added. The reaction mixture was stirred at 70 °C for 1 h and then diluted with CH_2_Cl_2_ (50 mL), washed with 1 M KHSO_4_ (50 mL), H_2_O (50 mL), and NaHCO_3_ (50 mL). Combined organic extracts were filtered through a cotton wool plug, concentrated, and dried in vacuo. The residue was purified by silica gel chromatography (gradient: 0%→15% CH_2_Cl_2_ in petroleum ether) to give known [28] silylated disaccharide **6** (27 mg, 63%).

(2) To a solution of CPP-glycoside **13** (91 mg, 0.075 mmol) in ClCH_2_CH_2_Cl (2 mL), *p*-TolSH (19 mg, 0.15 mmol) and BF_3_·Et_2_O (4 μL, 0.02 mmol) at −5 °C were added. Then, the temperature was raised to 0 °C and the reaction mixture was stirred at 0 °C for 2.5 h. Then, the reaction mixture was diluted with CH_2_Cl_2_ (50 mL) and washed with H_2_O (50 mL). The organic extract was filtered through a cotton wool plug, concentrated, and dried in vacuo. The residue was purified by silica gel chromatography (gradient: 0%→15% CH_2_Cl_2_ in petroleum ether) to give known [28] silylated disaccharide **6** (44 mg, 50%; 54% with respect to the reacted starting material).


**2,3,5-Tris-*O*-(triisopropylsilyl)-β**
**-d**
**-arabinofuranosyl-(1→**
**2)-1-*O*-acetyl-3,5-bis-*O*-(triisopropylsilyl)-α**
**-d-**
**arabinofuranose (10)**


To a solution of known arabinofuranose **9** [28] (48 mg, 0.045 mmol) in anhydrous Py (1 mL), Ac_2_O (1 mL) was added at 0 °C (ice–water bath). The reaction mixture was stirred at 20 °C for 24 h. The reaction was quenched by the addition of MeOH (1 μL) at 0 °C (ice–water bath), then concentrated under reduced pressure, co-evaporated with toluene (5×5 mL), and dried in vacuo to give acetate **10** (42 mg, 84%); *R*_f_ = 0.50 (light petroleum–CH_2_Cl_2_ 1:1); ${\left[\alpha \right]}_{D}^{28}$ −1.7 (c 2.1 in CHCl_3_); ^1^H NMR (300 MHz, CDCl_3_): δ 0.99–1.20 (m, 105H, 5 × ((CH_3_)_2_CH)_3_Si), 2.05 (s, 3H, CH_3_CO), 3.68 (dd, 1H, *J* 8.4 Hz, *J* 3.6 Hz, H-5^II^a), 3.71–3.82 (m, 2H, H-5^I^a, H-5^I^b), 3.82–3.99 (m, 2H, H-4^II^, H-5^II^b), 4.04 (dd, 1H, *J* 2.7 Hz, *J* 1.0 Hz, H-2^II^), 4.21 (d, 1H, *J* 0.8 Hz, H-2^I^), 4.25 (td, 1H, *J* 6.4 Hz, *J* 2.4 Hz, H-4^I^), 4.32 (d, 1H, *J* 1.1 Hz, H-3^II^), 4.56 (d, 1H, *J* 2.5 Hz, H-3^I^), 5.25 (d, 1H, *J* 2.7 Hz, H-1^II^), 6.16 (s, 1H, H-1^I^); ^13^C NMR (76 MHz, CDCl_3_): δ 12.0 (2 × ((CH_3_)_2_CH)_3_Si), 12.2 (2 × ((CH_3_)_2_CH)_3_Si), 12.3 ((CH_3_)_2_CH)_3_Si), 17.9 (2 × ((CH_3_)_2_CH)_3_Si), 17.97 (2 × ((CH_3_)_2_CH)_3_Si), 18.01 ((CH_3_)_2_CH)_3_Si), 18.0 ((CH_3_)_2_CH)_3_Si), 18.07 ((CH_3_)_2_CH)_3_Si), 18.11 (3 × ((CH_3_)_2_CH)_3_Si), 21.2 (CH_3_CO), 63.6 (C-5^II^), 64.2 (C-5^I^), 77.1 (C-3^I^), 78.0 (C-2^II^), 78.1 (C-3^II^), 86.0 (C-4^II^), 90.4 (C-2^I^), 90.8 (C-4^I^), 101.0 (C-1^I^), 105.3 (C-1^II^), 170.1 (CO); ^29^Si INEPT NMR (60 MHz, CDCl_3_) δ 13.67 (5^I^-*O*-TIPS, 5^II^-*O*-TIPS), 13.80 (5^I^-*O*-TIPS, 5^II^-*O*-TIPS), 13.96 (3^II^-*O*-TIPS), 14.02 (3^I^-*O*-TIPS), 14.29 (2^II^-*O*-TIPS); HRMS (ESI): *m/z* [M + NH_4_]^+^ Calcd for C_57_H_124_NO_10_Si_5_^+^ 1122.8066; Found: 1122.8063; [M + K]^+^ Calcd for C_57_H_120_KO_10_Si_5_^+^: 1143.7359; found: 1143.7357.


**4-(3-Chloropropoxy)phenyl 2,3,5-tris-*O*-(triisopropylsilyl)-β**
**-d**
**-arabinofuranosyl-(1→**
**2)-3,5-bis-*O*-(triisopropylsilyl)-α**
**-d-**
**arabinofuranoside (13)**


Known 4-(3-chloropropoxy)phenyl 3,5-*O*-(di-*tert*-butylsilylene)-2-*O*-[2-*O*-(triisopropylsilyl)-β-d-arabinofuranosyl]-α-d-arabinofuranoside **11** [28] (147 mg, 0.19 mmol) was dissolved in THF (2 mL), then 1 M TBAF in THF (0.77 mL, 0.77 mmol) was added to the reaction mixture at 0 °C. The reaction mixture was kept at 0 °C for 3 h. After that, the reaction mixture was concentrated under reduced pressure, co-evaporated with toluene (2 × 2 mL), and dried in vacuo. The residue was purified by silica gel chromatography (gradient: 5%→10% MeOH in CH_2_Cl_2_) to give 4-(3-chloropropoxy)phenyl 2-*O*-(β-d-arabinofuranosyl)-α-d-arabinofuranoside **12**, containing 3 mol. % of 4-(3-fluoropropoxy)phenyl 2-*O*-(β-d-arabinofuranosyl)-α-d-arabinofuranoside (72 mg, 81%) and 15 mol. % of *n*-Bu_4_N^+^ salts. Selected signals for **12**: ^1^H NMR (300 MHz, CD_3_OD): δ 2.18 (p, 2H, *J* 6.2 Hz, CH_2_), 3.62–3.86 (m, 7H), 3.94–4.09 (m, 5H), 4.13–4.20 (m, 1H, H-3^I^), 4.33–4.38 (m, 1H, H-2^I^), 5.03 (d, 1H, *J* 4.2 Hz, H-1^II^), 5.56 (d, 1H, *J* 2.5 Hz, H-1^I^), 6.79–6.92 (m, 2H, OC_6_H_4_O), 6.93–7.05 (m, 2H, OC_6_H_4_O); selected signals for minor 4-(3-fluoropropoxy)phenyl 2-*O*-(β-d-arabinofuranosyl)-α-d-arabinofuranoside: ^1^H NMR (300 MHz, CD_3_OD): 4.60 (dt, 2H, *J* 47.3 Hz, *J* 5.9 Hz, CH_2_F), ^19^F NMR (282 MHz, CD_3_OD): δ −223.86 (tt, ^2^J_H–F_ 47.3 Hz, ^3^J_H–F_ 25.4 Hz, CH_2_F). Then, *i*-Pr_3_SiOTf (0.42 mL, 1.57 mmol) was added to the solution of crude pentaol **12** (72 mg, 0.16 mmol) in 2,4,6-collidine (1.5 mL) at 20 °C. The reaction mixture was stirred at 80 °C for 20 h and then diluted with CH_2_Cl_2_ (30 mL), washed with 1 M KHSO_4_ (3 × 30 mL), and satd aq NaHCO_3_ (3 × 30 mL). Organic extracts were filtered, concentrated under reduced pressure, and purified by silica gel chromatography (gradient: petroleum ether–EtOAc, 0%→6%) to give silylated disaccharide **13**, containing 3% 4-(3-fluoropropoxy)phenyl derivative (155 mg, 80%). *R*_f_ = 0.80 (light petroleum–EtOAc 10:1); ${\left[\alpha \right]}_{D}^{22}$ +21.1 (c 0.25 in CHCl_3_); ^1^H NMR (600 MHz, CDCl_3_): δ 1.03–1.22 (m, 105H, ((CH_3_)_2_CH)_3_Si), 2.15 (dt, 1H, *J* 25.8 Hz, *J* 6.0 Hz, CH_2_CH_2_F), 2.22 (p, 2H, *J* 6.1 Hz, CH_2_), 3.71 (dd, 1H, *J* 9.3 Hz, *J* 4.5 Hz, H-5^II^a), 3.75 (t, 2H, *J* 6.4 Hz, CH_2_Cl), 3.78 (dd, 1H, *J* 10.7 Hz, *J* 5.6 Hz, H-5^I^a), 3.87 (dd, 1H, *J* 10.7 Hz, *J* 5.3 Hz, H-5^I^b), 3.88 (dd, 1H, *J* 10.4 Hz, *J* 4.5 Hz, H-4^II^), 3.94 (dd, 1H, *J* 10.4 Hz, *J* 9.3 Hz, H-5^II^b), 4.01 (d, 1H, *J* 2.9 Hz, H-2^II^), 4.07 (t, 2H, *J* 5.8 Hz, CH_2_O), 4.20 (td, 1H, *J* 5.5 Hz, *J* 4.4 Hz, H-4^I^), 4.33 (d, 1H, *J* 1.1 Hz, H-3^II^), 4.41 (dd, 1H, *J* 2.2 Hz, *J* 1.0 Hz, H-2^I^), 4.52 (dd, 1H, *J* 4.5 Hz, *J* 2.2 Hz, H-3^I^), 4.65 (dt, 1H, *J* 47.1 Hz, *J* 5.8 Hz, CH_2_F), 5.29 (d, 1H, *J* 2.8 Hz, H-1^II^), 5.53 (d, 1H, *J* 0.8 Hz, H-1^I^), 6.78–6.84 (m, 2H, OC_6_H_4_O (H-3, H-5)), 6.94–7.00 (m, 2H, OC_6_H_4_O (H-2, H-6)); ^13^C NMR (151 MHz, CDCl_3_): δ 11.98 (((CH_3_)_2_CH)_3_Si), 12.03 (((CH_3_)_2_CH)_3_Si), 12.30 (((CH_3_)_2_CH)_3_Si), 12.32 (((CH_3_)_2_CH)_3_Si), 12.4 (((CH_3_)_2_CH)_3_Si), 17.96 (((CH_3_)_2_CH)_3_Si), 17.98 (((CH_3_)_2_CH)_3_Si), 18.08 (((CH_3_)_2_CH)_3_Si), 18.11 (((CH_3_)_2_CH)_3_Si), 18.14 (((CH_3_)_2_CH)_3_Si), 18.15 (((CH_3_)_2_CH)_3_Si), 32.4 (CH_2_), 41.6 (CH_2_Cl), 63.7 (C-5^II^), 63.8 (C-5^I^), 64.9 (CH_2_O), 77.1 (C-3^I^), 78.1 (C-2^II^), 78.3 (C-3^II^), 85.9 (C-4^II^), 87.9 (C-4^I^), 91.2 (C-2^I^), 104.8 (C-1^II^), 105.6 (C-1^I^), 115.3 (OC_6_H_4_O (C-3, C-5)), 118.1 (OC_6_H_4_O (C-2, C-6)), 151.6 (OC_6_H_4_O (C-1)), 153.7 (OC_6_H_4_O (C-4)); ^29^Si INEPT NMR (60 MHz, CDCl_3_): δ 13.36 (3^I^-*O*-TIPS), 13.65 (5^I^-*O*-TIPS, 5^II^-*O*-TIPS), 13.70 (3^II^-*O*-TIPS), 14.05 (2^II^-*O*-TIPS); HRMS (ESI): *m*/*z* [M + NH_4_]^+^ Calcd for C_64_H_131_ClNO_10_Si_5_^+^ 1248.8302; found: 1248.8296.


**4-(2-Chloroethoxy)phenyl 2,3,5-tris-*O*-(triisopropylsilyl)-β**
**-d**
**-arabinofuranosyl-(1→**
**2)-3,5-bis-*O*-(triisopropylsilyl)-α**
**-d-**
**arabinofuranosyl-(1→**
**5)-2-*O*-benzoyl-α**
**-d-**
**arabinofuranosyl-(1→**
**5)-2,3-di-*O*-benzoyl-α**
**-d-**
**arabinofuranoside (14)**


A mixture of disaccharide thioglycoside **6** (82 mg, 0.07 mmol) and disaccharide diol **8** [28] (19 mg, 0.25 mmol) was dried in vacuo for 2 h, then anhydrous CH_2_Cl_2_ (2 mL) was added under argon. Freshly activated powdered MS 4 Å (200 mg) (100 mg per 1 mL of solvent) was added under argon to the resulting solution. The suspension was stirred under argon at ~22 °C for 1 h, cooled to −60 °C, then NIS (16 mg, 0.7 mmol) and, after 10 min, AgOTf (1 mg) was added. Then, the temperature was allowed to rise slowly to −20 °C and was kept at −20 °C for 1 h. Then, the reaction was quenched by the addition of satd aq NaHCO_3_ (50 μL), diluted with CH_2_Cl_2_ (15 mL), and filtered through a Celite pad. The solids were washed with CH_2_Cl_2_ (5 × 10 mL), and the filtrate was washed with a mixture of satd aq Na_2_S_2_O_3_ (50 mL) and satd aq NaHCO_3_ (50 mL). The aqueous layer was extracted with CH_2_Cl_2_ (2 × 5 mL). The combined organic extracts were filtered through a cotton wool plug, concentrated and dried in vacuo, then dissolved in toluene (2 mL) and subjected to chromatography on Bio-Beads S-X1 in toluene. The fractions eluted just after the void volume were collected, concentrated under reduced pressure, and the residue was purified by silica gel chromatography (gradient: EtOAc in petroleum ether, 0%→30%) to give α-linked tetrasaccharide **14** (36 mg, 80%). *R*_f_ = 0.20 (light petroleum–EtOAc 10:1); ^1^H NMR (600 MHz, CDCl_3_): δ 0.89 –1.14 (m, 105H, 5 × ((CH_3_)_2_CH)_3_Si), 3.43 (d, 1H, *J* 4.1 Hz, HO-3^II^), 3.66 (dd, 1H, *J* 9.5 Hz, *J* 4.6 Hz, H-5^IV^a), 3.68 (dd, 1H, *J* 11.1 Hz, *J* 3.9 Hz, H-5^II^a), 3.73 (dd, 1H, *J* 10.4 Hz, *J* 6.4 Hz, H-5^III^a), 3.77–3.80 (m, 1H, H-5^III^b), 3.79 (t, 2H, *J* 5.9 Hz, CH_2_Cl), 3.82 (dd, 1H, *J* 10.3 Hz, *J* 4.6 Hz, H-4^IV^), 3.90 (dd, 1H, *J* 10.2 Hz, *J* 9.4 Hz, H-5^IV^b), 3.92 (dd, 1H, *J* 11.1 Hz, *J* 3.7 Hz, H-5^I^a), 3.98 (d, 1H, *J* 3.0 Hz, H-2^IV^), 4.00 (dd, 1H, *J* 11.2 Hz, *J* 3.5 Hz, H-5^II^b), 4.07 (td, 1H, *J* 6.0 Hz, *J* 3.6 Hz, H-4^III^), 4.17 (d, 1H, *J* 1.2 Hz, H-2^III^), 4.19 (t, 2H, *J* 6.0 Hz, CH_2_O), 4.21 (dd, 1H, *J* 11.0 Hz, *J* 4.5 Hz, H-5^I^b), 4.29 (d, 1H, *J* 1.1 Hz, H-3^IV^), 4.31 (dt, 1H, *J* 7.1 Hz, *J* 3.5 Hz, H-3^II^), 4.34–4.38 (m, 2H, H-4^II^, H-3^III^), 4.58–4.63 (m, 1H, H-4^I^), 5.06 (s, 1H, H-1^III^), 5.17 (dd, 1H, *J* 3.7 Hz, *J* 1.4 Hz, H-2^II^), 5.20 (d, 1H, *J* 2.6 Hz, H-1^IV^), 5.37 (d, 1H, *J* 1.4 Hz, H-1^II^), 5.68 (dd, 1H, *J* 5.0 Hz, *J* 1.8 Hz, H-3^I^), 5.75 (d, 1H, *J* 1.7 Hz, H-2^I^), 5.83 (s, 1H, H-1^I^), 6.84–6.91 (m, 2H, OC_6_H_4_O (H-3, H-5)), 7.04–7.11 (m, 2H, OC_6_H_4_O (H-2, H-6)), 7.41–7.51 (m, 6H, 3 × PhCO (H-3, H-5)), 7.55–7.64 (m, 3H, 3 × PhCO (H-4)), 7.99–8.04 (m, 2H, 2^II^-*O*-PhCO (H-2, H-6)), 8.07–8.11 (m, 2H, 2^I^-*O*-PhCO (H-2, H-6)), 8.09–8.14 (m, 2H, 3^I^-*O*-PhCO (H-2, H-6)); ^13^C NMR (151 MHz, CDCl_3_): δ 11.9 (((CH_3_)_2_CH)_3_Si), 12.0 (((CH_3_)_2_CH)_3_Si), 12.1 (((CH_3_)_2_CH)_3_Si), 12.2 (((CH_3_)_2_CH)_3_Si), 12.3 (((CH_3_)_2_CH)_3_Si), 17.9 (2 × ((CH_3_)_2_CH)_3_Si), 17.95 (3 × ((CH_3_)_2_CH)_3_Si), 17.97 (((CH_3_)_2_CH)_3_Si), 18.06 (((CH_3_)_2_CH)_3_Si), 18.09 (((CH_3_)_2_CH)_3_Si), 18.13 (((CH_3_)_2_CH)_3_Si), 18.14 (((CH_3_)_2_CH)_3_Si), 41.9 (CH_2_Cl), 63.7 (C-5^IV^), 64.4 (C-5^II^), 64.6 (C-5^III^), 66.0 (C-5^I^), 68.7 (CH_2_O), 76.1 (C-3^II^), 77.3 (C-3^I^), 77.9 (C-3^III^), 78.0 (C-2^IV^), 78.1 (C-3^IV^), 82.1 (C-2^I^, C-4^II^), 82.5 (C-4^I^), 85.8 (C-4^IV^), 86.9 (C-2^II^), 88.2 (C-4^III^), 90.8 (C-2^III^), 104.8 (C-1^I^), 105.0 (C-1^IV^), 105.4 (C-1^II^), 106.3 (C-1^III^), 115.8 (OC_6_H_4_O (C-3, C-5)), 118.3 (OC_6_H_4_O (C-2, C-6)), 128.4 (PhCO (C-3, C-5)), 128.5 (PhCO (C-3, C-5)), 128.6 (PhCO (C-3, C-5)), 128.9 (PhCO (C-1)), 129.1 (PhCO (C-1)), 129.3 (PhCO (C-1)), 129.87 (PhCO (C-2, C-6)), 129.92 (PhCO (C-2, C-6)), 129.94 (PhCO (C-2, C-6)), 133.4 (PhCO (C-4)), 133.5 (PhCO (C-4)), 133.6 (PhCO (C-4)), 150.7 (OC_6_H_4_O (C-1)), 153.7 (OC_6_H_4_O (C-4)), 165.5 (2^I^-*O*-PhCO), 165.7 (3^I^-*O*-PhCO), 166.8 (2^II^-*O*-PhCO); ^29^Si INEPT NMR (60 MHz, CDCl_3_): δ 13.54, 13.59, 13.70, 13.80, 13.89; HRMS (ESI): *m*/*z* [M + NH_4_]^+^ Calcd for C_94_H_193_ClNO_21_Si_5_^+^ 1810.9777; found: 1810.9764; [M + K]^+^ Calcd for C_94_H_153_ClKO_21_Si_5_^+^: 1831.9071; found: 1831.9063.


**2,3,5-Tris-*O*-(triisopropylsilyl)-β**
**-d**
**-arabinofuranosyl-(1→**
**2)-3,5-bis-*O*-(triisopropylsilyl)-α**
**-d-**
**arabinofuranosyl-(1→**
**3)-[2,3,5-tris-*O*-(triisopropylsilyl)-β**
**-d**
**-arabinofuranosyl-(1→**
**2)-3,5-bis-*O*-(triisopropylsilyl)-α**
**-d-**
**arabinofuranosyl-(1→**
**5)]-2-*O*-benzoyl-α**
**-d-**
**arabinofuranosyl-(1→**
**5)-2,3-di-*O*-benzoyl**
**-d-**
**arabinofuranose (15) and 2,3,5-tris-*O*-(triisopropylsilyl)-β**
**-d**
**-arabinofuranosyl-(1→**
**2)-3,5-bis-*O*-(triisopropylsilyl)-α**
**-d-**
**arabinofuranosyl-(1→**
**3)-[2,3,5-tris-*O*-(triisopropylsilyl)-β**
**-d**
**-arabinofuranosyl-(1→**
**2)-3,5-bis-*O*-(triisopropylsilyl)-α**
**-d-**
**arabinofuranosyl-(1→**
**5)]-2-*O*-benzoyl-α**
**-d-**
**arabinofuranosyl-(1→**
**5)-1,3-di-*O*-benzoyl-β**
**-d-**
**arabinofuranose (16)**


CEP glycoside **2** (44 mg, 0.016 mmol) was dissolved in CH_2_Cl_2_ (2 mL) and MeCN (1 mL), then H_2_O (0.1 mL) was added at 0 °C, followed by (NH_4_)_2_Ce(NO_3_)_6_ (52.6 mg, 0.096 mmol). The reaction mixture was stirred at 0 °C for 4 h. Then, Na_2_SO_3_ (110 mg) and H_2_O (1 mL) were added to the reaction mixture at 0 °C, and the reaction mixture was kept at 0 °C for 16 h. Then, the reaction mixture was diluted with CH_2_Cl_2_ (50 mL), washed with H_2_O (50 mL), concentrated under reduced pressure and purified by silica gel column chromatography (gradient: petroleum ether–EtOAc, 3%→8%) to give hemiacetal **15** (29 mg, 70%) as a mixture of anomers **15**, and contaminated with the product **16** of migration of benzoyl group from O-2 to O-1. The ratio of **15**α:**15**β:**16** = 1:0.45:0.28 according to NMR; *R*_f_ = 0.50 (light petroleum–EtOAc, 8.5:1.5); selected signals: ^1^H NMR (300 MHz, CDCl_3_): δ 0.75–1.18 (m, 363H, ((CH_3_)_2_CH)_3_Si), 3.40 (s, 1H, HO-1^I^α), 4.65 (ddd, 1H, *J* 6.3 Hz, *J* 5.2 Hz, *J* 3.3 Hz, H-4^I^α), 5.05 (s, 1H, H-1^III^ of **16**), 5.10 (s, 1H, H-1^III^β), 5.10 (s, 1H, H-1^III^α), 5.16 (s, 1H, H-1^V^β), 5.17–5.18 (m, 2H, H-1^II^β, H-1^II^ or H-1^V^ of **16**), 5.31 (d, 1H, *J* 3.2 Hz, H-2^II^β), 5.35 (d, 1H, *J* 2.0 Hz, H-2^II^α), 5.40–5.48 (m, 2H, H-3^I^α, H-3^I^ of **16**), 5.51 (d, 1H, *J* 2.4 Hz, H-2^I^α), 5.56 (dd, 1H, *J* 6.2 Hz, *J* 4.7 Hz, H-2^I^β), 5.63 (s, 1H, H-1^I^α), 5.72 (dd, 1H, *J* 5.5 Hz, *J* 5.5 Hz, H-1^I^β), 5.86 (dd, 1H, *J* 6.2 Hz, *J* 5.0 Hz, H-3^I^β), 6.60 (d, 1H, *J* 4.5 Hz, H-1^I^ of **16**), 7.33–7.66 (m, 16H, PhCO (H-3, H-4, H-5)), 7.92–8.19 (m, 11H, PhCO (H-2, H-6)); ^13^C NMR (151 MHz, CDCl_3_): δ 11.97 (((CH_3_)_2_CH)_3_Si), 11.99 (((CH_3_)_2_CH)_3_Si), 12.02 (((CH_3_)_2_CH)_3_Si), 12.03 (((CH_3_)_2_CH)_3_Si), 12.2 (((CH_3_)_2_CH)_3_Si), 12.28 (((CH_3_)_2_CH)_3_Si), 12.31 (((CH_3_)_2_CH)_3_Si), 12.36 (((CH_3_)_2_CH)_3_Si), 12.39 (((CH_3_)_2_CH)_3_Si), 17.95 (((CH_3_)_2_CH)_3_Si), 17.97 (((CH_3_)_2_CH)_3_Si), 17.99 (((CH_3_)_2_CH)_3_Si), 18.01 (((CH_3_)_2_CH)_3_Si), 18.03 (((CH_3_)_2_CH)_3_Si), 18.05 (((CH_3_)_2_CH)_3_Si), 18.08 (((CH_3_)_2_CH)_3_Si), 18.09 (((CH_3_)_2_CH)_3_Si), 18.11 (((CH_3_)_2_CH)_3_Si), 18.13 (((CH_3_)_2_CH)_3_Si), 18.14 (((CH_3_)_2_CH)_3_Si), 18.17 (((CH_3_)_2_CH)_3_Si), 18.18 (((CH_3_)_2_CH)_3_Si), 18.20 (((CH_3_)_2_CH)_3_Si), 63.7 (C-5), 63.76 (2 × C-5), 63.82 (C-5), 64.16 (C-5), 64.18 (C-5), 64.5 (C-5), 64.7 (C-5), 65.3 (C-5^II^ minor), 65.4 (C-5^II^α), 66.0 (C-5^II^β), 66.6 (C-5^I^α), 67.4 (C-5^I^ of **16**), 67.6 (C-5^I^β), 76.5 (C-3^I^β), 76.7 (C-2^I^ of **16**), 77.4, 77.5, 77.7, 78.08, 78.14, 78.19, 78.21, 78.23, 78.25, 78.28, 78.32, 78.37, 78.43, 78.47, 78.52, 79.7, 80.06 (C-3^II^β), 80.12, 80.6, 80.7 (C-3^II^α), 81.0 (C-3^I^ of **16**), 81.1, 81.8 (C-4^I^α), 82.1, 82.2, 82.6 (C-2^I^α), 83.6 (C-2^II^α), 83.7 (C-2^II^ of **16**), 84.1 (C-2^II^β), 85.7 (C-4^IV^, C-4^VI^), 85.7 (C-4^IV^, C-4^VI^), 85.6 (C-4^IV^, C-4^VI^), 85.7 (C-4^IV^, C-4^VI^), 85.8 (C-4^IV^, C-4^VI^), 87.5 (C-4^III^), 87.54 (C-4^III^), 87.59 (C-4^III^ of **16**), 87.8 (C-4^V^), 88.3 (C-4^V^), 88.4 (C-4^V^ of **16**), 90.5 (C-2^III^), 90.70 (C-2^III^), 90.74 (C-2^III^ of **16**), 91.2 (C-2^V^), 91.4 (C-2^V^ of **16**), 91.6 (C-2^V^), 95.4 (C-1^I^β), 97.0 (C-1^I^ of **16**), 100.9 (C-1^I^α), 104.7 (C-1), 104.75 (C-1), 104.78 (C-1 minor), 105.2 (C-1), 105.5 (C-1), 105.48 (C-1 of **16**), 105.93 (C-1), 105.94 (C-1 of **16**), 105.98 (C-1), 106.04 (C-1), 106.2 (C-1), 106.4 (C-1 of **16**), 106.69 (C-1^III^ of **16**), 106.73 (C-1^III^β), 106.8 (C-1^III^α), 128.20 (PhCO (C-3, C-5) of **16**), 128.23 (PhCO (C-3, C-5)), 128.27 (PhCO (C-3, C-5)), 128.30 (PhCO (C-3, C-5)), 128.37 (2 × PhCO (C-3, C-5)), 128.44 (PhCO (C-3, C-5) of **16**), 128.5 (PhCO (C-3, C-5)), 129.3 (PhCO (C-1)), 129.4 (PhCO (C-1)), 129.48 (PhCO (C-1)), 129.49 (PhCO (C-1)), 129.54 (PhCO (C-1)), 129.8 (PhCO (C-1)), 129.85 (PhCO (C-2, C-6)), 129.86 (PhCO (C-2, C-6)), 129.92 (PhCO (C-2, C-6) of **16**), 129.94 (PhCO (C-2, C-6)), 130.0 (2 × PhCO (C-2, C-6)), 130.1 (PhCO (C-2, C-6)), 132.9 (PhCO (C-4)), 133.07 (PhCO (C-4)), 133.12 (PhCO (C-4)), 133.2 (PhCO (C-4)), 133.27 (PhCO (C-4)), 133.29 (PhCO (C-4)), 133.5 (PhCO (C-4) of **16**), 165.16 (PhCO of **16**), 165.20 (PhCO), 165.58 (PhCO), 165.59 (PhCO), 165.7 (PhCO), 165.8 (2 × PhCO); HRMS (ESI): *m*/*z* [M + NH_4_]^+^ Calcd for C_141_H_266_NO_28_Si_10_^+^ 2701.7109; found: 2701.7093.


**4-(2-Chloroethoxy)phenyl 2,3,5-tris-*O*-(triisopropylsilyl)-β**
**-d**
**-arabinofuranosyl-(1→**
**2)-3,5-bis-*O*-(triisopropylsilyl)-α**
**-d-**
**arabinofuranosyl-(1→**
**3)-[2,3,5-tris-*O*-(triisopropylsilyl)-β**
**-d**
**-arabinofuranosyl-(1→**
**2)-3,5-bis-*O*-(triisopropylsilyl)-α**
**-d-**
**arabinofuranosyl-(1→**
**5)]-2-*O*-benzoyl-α**
**-d-**
**arabinofuranosyl-(1→**
**5)-2,3-di-*O*-benzoyl-α**
**-d-**
**arabinofuranosyl-(1→**
**5)-2,3-di-*O*-benzoyl-α-d-arabinofuranosyl-(1→**
**5)-2,3-di-*O*-benzoyl-α-d-arabinofuranosyl-(1→**
**5)-2,3-di-*O*-benzoyl-α-d-arabinofuranosyl-(1→**
**5)-2,3-di-*O*-benzoyl-α-d-arabinofuranoside (18)**


A mixture of hexasaccharide imidate **5** (32 mg, 0.011 mmol) and known tetrasaccharide glycosyl acceptor **17** [33] (21 mg, 0.013 mmol) was dried *in vacuo* for 2 h, then anhydrous CH_2_Cl_2_ (2 mL) was added under argon. Freshly activated powdered MS 4 Å (200 g) (100 mg per 1 mL of solvent) was added under argon to the resulting solution. The suspension was stirred under argon at ~22 °C for 1 h, then cooled to −60 °C, followed by the addition of TfOH (1 μL). Then, the temperature was allowed to rise during 1 h to −40 °C and this temperature was kept for 1 h. After an additional 20 min, the reaction was quenched by the addition of Py (50 µL). The reaction mixture was diluted with CH_2_Cl_2_ (15 mL) and filtered through a Celite pad. The solids were washed with CH_2_Cl_2_ (5 × 10 mL), and the filtrate was washed with satd aq NaHCO_3_ (20 mL). The aqueous layer was extracted with CH_2_Cl_2_ (2 × 5 mL). The combined organic layer was filtered through a cotton wool plug, concentrated under reduced pressure, the residue was dried in vacuo, then dissolved in toluene (2 mL) and subjected to chromatography on Bio-Beads S-X1 in toluene. The fractions eluted just after the void volume were collected and concentrated under reduced pressure to give decasaccharide **18** (34 mg, 72%; 51% over 3 steps starting from **2**). *R*_f_ = 0.28 (light petroleum–EtOAc 6:1); ${\left[\alpha \right]}_{D}^{23}$ +31.9 (c 1.13 in CHCl_3_); ^1^H NMR (600 MHz, CDCl_3_): δ 0.91–1.16 (m, 210H, 10 × ((CH_3_)_2_CH)_3_Si), 3.63 (dd, 1H, *J* 9.3 Hz, *J* 4.4 Hz, H-5^VIII^a or H-5^X^a), 3.65 (dd, 1H, *J* 10.7 Hz, *J* 5.5 Hz, H-5^IX^a), 3.78 (t, 2H, *J* 5.9 Hz, CH_2_Cl), 3.69–3.98 (m, 16H, H-5^X^a or H-5^VIII^a, H-5^I-VII^a, H-5^VI-X^b, H-4^VIII-X^), 4.02–4.04 (m, 2H, H-2^VIII^, H-2^X^), 4.06 (d, 1H, *J* 1.2 Hz, H-2^IX^), 4.08–4.12 (m, 2H, H-4^VII^, H-5^V^b), 4.13 (s, 1H, H-2^VII^), 4.16 (t, 2H, *J* 5.9 Hz, CH_2_O), 4.14–4.24 (m, 5H, H-5^I-IV^, H-3^VI^), 4.26–4.28 (m, 1H, H-4^VI^), 4.29 (s, 1H, H-3^X^), 4.32 (s, 1H, H-3^VIII^), 4.34 (d, 1H, *J* 3.9 Hz, H-3^VII^), 4.45 (d, 1H, *J* 3.4 Hz, H-3^IX^), 4.55–4.64 (m, 5H, H-4^I-V^), 5.10 (s, 1H, H-1^VII^), 5.15 (d, 1H, *J* 2.6 Hz, H-1^X^), 5.23 (s, 1H, H-1^IX^), 5.24 (d, 1H, *J* 2.8 Hz, H-1^VIII^), 5.29 (s, 1H, H-1^VI^), 5.37 (s, 2H, 2 × H-1), 5.38–5.41 (m, 3H, H-2^VI^, 2 × H-1), 5.50 (d, 1H, *J* 4.4 Hz, H-3^V^), 5.58 (d, 1H, *J* 1.1 Hz, H-2^V^), 5.62–5.67 (m, 6H, H-3^II-IV^, H-2^II-IV^), 5.75–5.79 (m, 2H, H-3^I^, H-2^I^), 5.82 (s, 1H, H-1^I^), 6.80–6.86 (m, 2H, OC_6_H_4_O (H-3, H-5)), 7.02–7.08 (m, 2H, OC_6_H_4_O (H-2, H-6)), 7.18–7.27 (m, 8H, 4 × PhCO (H-3, H-5)), 7.34–7.55 (m, 24H, 7 × PhCO (H-3, H-5), 10 × PhCO (H-4)), 7.55–7.61 (m, 1H, PhCO (H-4)), 7.84–7.92 (m, 8H, 4 × PhCO (H-2, H-6)), 7.97–8.06 (m, 10H, 5 × PhCO (H-2, H-6)), 8.06–8.11 (m, 2H, PhCO (H-2, H-6)), 8.09–8.14 (m, 2H, PhCO (H-2, H-6));^13^C NMR (151 MHz, CDCl_3_): δ 11.97 (((CH_3_)_2_CH)_3_Si), 11.99 (((CH_3_)_2_CH)_3_Si), 12.01 (((CH_3_)_2_CH)_3_Si), 12.1 (((CH_3_)_2_CH)_3_Si), 12.2 (((CH_3_)_2_CH)_3_Si), 12.27 (2 × ((CH_3_)_2_CH)_3_Si), 12.29 (2 × ((CH_3_)_2_CH)_3_Si), 12.4 (((CH_3_)_2_CH)_3_Si), 17.94 (((CH_3_)_2_CH)_3_Si), 17.96 (((CH_3_)_2_CH)_3_Si), 17.97 (((CH_3_)_2_CH)_3_Si), 18.02 (((CH_3_)_2_CH)_3_Si), 18.04 (((CH_3_)_2_CH)_3_Si), 18.06 (((CH_3_)_2_CH)_3_Si), 18.09 (((CH_3_)_2_CH)_3_Si), 18.13 (((CH_3_)_2_CH)_3_Si), 18.19 (((CH_3_)_2_CH)_3_Si), 18.20 (((CH_3_)_2_CH)_3_Si), 41.9 (CH_2_Cl), 63.8 (C-5^VIII^, C-5^X^), 63.9 (C-5^VIII^, C-5^X^), 64.3 (C-5^IX^), 64.9 (C-5^VII^), 65.2 (C-5^VI^), 65.3 (C-5^V^), 65.8 (2C, C-5^I^, C-5^II^, C-5^III^, C-5^IV^), 65.86 (C-5^I^, C-5^II^, C-5^III^, C-5^IV^), 65.91 (C-5^I^, C-5^II^, C-5^III^, C-5^IV^), 68.8 (CH_2_O), 77.2 (C-3^I^), 77.2 (2C, C-3^II^, C-3^III^, C-3^IV^, C-3^V^), 77.3 (C-3^II^, C-3^III^, C-3^IV^, C-3^V^), 77.6 (C-3^II^, C-3^III^, C-3^IV^, C-3^V^), 77.7 (C-3^IX^), 78.2 (C-2^VIII^ or C-2^X^), 78.3 (C-3^X^), 78.4(C-2^VIII^ or C-2^X^), 78.5 (2C, C-3^VII^, C-3^VIII^), 81.0 (C-3^VI^), 81.5 (2C, C-2^II^, C-2^III^, C-2^IV^, C-2^V^), 81.6 (2C, C-2^II^, C-2^III^, C-2^IV^, C-2^V^), 81.9 (C-2^I^), 82.2 (3C, C-4^II^, C-4^III^, C-4^IV^, C-4^V^, C-4^VI^), 82.26 (C-4^II^, C-4^III^, C-4^IV^, C-4^V^, C-4^VI^), 82.30 (C-4^II^, C-4^III^, C-4^IV^, C-4^V^, C-4^VI^), 82.8 (C-4^I^), 84.0 (C-2^VI^), 85.7 (C-4^VIII^, C-4^X^), 85.6 (C-4^VIII^, C-4^X^), 87.7 (C-4^VII^), 88.6 (C-4^IX^), 90.6 (C-2^VII^), 91.4 (C-2^IX^), 104.8 (C-1^VIII^), 104.9 (C-1^I^), 105.5 (C-1^X^), 105.85 (C-1^VI^), 105.89 (C-1), 106.0 (3C, C-1^II^, C-1^III^, C-1^IV^), 106.2 (C-1^IX^), 106.9 (C-1^VII^), 115.9 (OC_6_H_4_O (C-3, C-5)), 118.3 (OC_6_H_4_O (C-2, C-6)), 128.1 (PhCO (C-3, C-5)), 128.15 (PhCO (C-3, C-5)), 128.17 (PhCO (C-3, C-5)), 128.20 (PhCO (C-3, C-5)), 128.3 (PhCO (C-3, C-5)), 128.4 (PhCO (C-3, C-5)), 128.6 (PhCO (C-3, C-5)), 128.49 (3 × PhCO (C-3, C-5)), 128.54 (PhCO (C-3, C-5)), 129.1–129.29 (PhCO (C-1)), 129.29 (PhCO (C-1)), 129.36 (PhCO (C-1)), 129.41 (PhCO (C-1)), 129.5 (PhCO (C-1)), 129.78 (3 × PhCO (C-2, C-6)), 129.81 (2 × PhCO (C-2, C-6)), 129.83 (3 × PhCO (C-2, C-6)), 129.86 (PhCO (C-2, C-6)), 129.92 (PhCO (C-2, C-6)), 130.0 (PhCO (C-2, C-6)), 132.75 (PhCO (C-4)), 132.82 (PhCO (C-4)), 132.9 (PhCO (C-4)), 133.0 (PhCO (C-4)), 133.1 (2 × PhCO (C-4)), 133.2 (PhCO (C-4)), 133.3 (PhCO (C-4)), 133.35 (PhCO (C-4)), 133.44 (PhCO (C-4)), 133.5 (PhCO (C-4)), 150.8 (OC_6_H_4_O (C-1)), 153.7 (OC_6_H_4_O (C-4)), 165.0 (PhCO), 165.2 (PhCO), 165.2 (PhCO), 165.2 (PhCO), 165.2 (PhCO), 165.3 (PhCO), 165.37 (PhCO), 165.44 (PhCO), 165.5 (PhCO), 165.57 (PhCO), 165.64 (PhCO); ^29^Si INEPT NMR (60 MHz, CDCl_3_): δ 12.30 (3^VII^-*O*-TIPS), 12.84 (3^IX^-*O*-TIPS), 13.21 (5-*O*-TIPS), 13.44 (5-*O*-TIPS), 13.49 (3^X^-*O*-TIPS, 5-*O*-TIPS), 13.50 (3^X^-*O*-TIPS, 5-*O*-TIPS), 13.54 (5-*O*-TIPS), 13.76 (3^VIII^-*O*-TIPS), 13.82 (2^VIII^-*O*-TIPS, 2^X^-*O*-TIPS), 13.86 (2^VIII^-*O*-TIPS, 2^X^-*O*-TIPS); HRMS (ESI): *m*/*z* [M + 2NH_4_]^2+^ Calcd for C_225_H_341_ClN_2_O_53_Si_10_^2+^ 2117.0710; found: 2117.0702.


**4-(2-Azidoethoxy)phenyl 2,3,5-tris-*O*-(triisopropylsilyl)-β**
**-d**
**-arabinofuranosyl-(1→**
**2)-3,5-bis-*O*-(triisopropylsilyl)-α**
**-d-**
**arabinofuranosyl-(1→**
**3)-[2,3,5-tris-*O*-(triisopropylsilyl)-β**
**-d**
**-arabinofuranosyl-(1→**
**2)-3,5-bis-*O*-(triisopropylsilyl)-α**
**-d-**
**arabinofuranosyl-(1→**
**5)]-2-*O*-benzoyl-α**
**-d-**
**arabinofuranosyl-(1→**
**5)-2,3-di-*O*-benzoyl-α**
**-d-**
**arabinofuranosyl-(1→**
**5)-2,3-di-*O*-benzoyl-α-d-arabinofuranosyl-(1→**
**5)-2,3-di-*O*-benzoyl-α-d-arabinofuranosyl-(1→**
**5)-2,3-di-*O*-benzoyl-α-d-arabinofuranosyl-(1→**
**5)-2,3-di-*O*-benzoyl-α-d-arabinofuranoside (19)**


A mixture of decasaccharide CEP glycoside **18** (30 mg, 0.007 mmol), NaN_3_ (3 mg, 0.042 mmol), and 18-crown-6 (2 mg, 0.006 mmol) in DMF (1 mL) was stirred at 80 °C for 16 h. The reaction mixture was concentrated under reduced pressure, co-evaporated with toluene (2×2 mL), and dried in vacuo. The residue was dissolved in EtOAc (50 mL), washed with water (50 mL), and the aqueous layer was extracted with EtOAc (5 mL). The combined organic extracts were filtered through a cotton wool plug, concentrated under reduced pressure, and purified by silica gel column chromatography (gradient: EtOAc in petroleum ether, 2%→20%) to give azide **19** (25 mg, 84%); ${\left[\alpha \right]}_{D}^{22}$ +29.7 (c 2.55 in CHCl_3_); ^1^H NMR (600 MHz, CDCl_3_): δ 0.91–1.13 (m, 210H, 10 × ((CH_3_)_2_CH)_3_Si), 3.56 (t, 2H, *J* 5.0 Hz, CH_2_N_3_), 3.63 (dd, 1H, *J* 9.3 Hz, *J* 4.4 Hz, H-5^VIII^a or H-5^X^a), 3.66 (dd, 1H, *J* 10.5 Hz, *J* 5.5 Hz, H-5^IX^a), 3.69–3.98 (m, 16H, H-5^X^a or H-5^VIII^a, H-5^I-VII^a, H-5^VI-X^b, H-4^VIII-X^), 4.03 (d, 1H, *J* 2.8 Hz, H-2^VIII^), 4.04 (d, 1H, *J* 2.7 Hz, H-2^X^), 4.07 (d, 1H, *J* 1.4 Hz, H-2^IX^), 4.08 (t, 2H, *J* 5.1 Hz, CH_2_O), 4.07–4.13 (m, 2H, H-4^VII^, H-5^V^b), 4.13 (d, 1H, *J* 1.2 Hz, H-2^VII^), 4.14–4.24 (m, 5H, H-5^I-IV^b, H-3^VI^), 4.27 (ddd, 1H, *J* 7.2 Hz, *J* 5.0 Hz, *J* 1.9 Hz, H-4^VI^), 4.29 (d, 1H, *J* 1.0 Hz, H-3^X^), 4.32 (d, 1H, *J* 0.9 Hz, H-3^VIII^), 4.34 (dt, 1H, *J* 4.0 Hz, *J* 0.8 Hz, H-3^VII^), 4.45 (dd, 1H, *J* 3.3 Hz, *J* 1.6 Hz, H-3^IX^), 4.54–4.64 (m, 5H, H-4^I-V^), 5.09 (s, 1H, H-1^VII^), 5.16 (d, 1H, *J* 2.7 Hz, H-1^X^), 5.23 (s, 1H, H-1^IX^), 5.24 (d, 1H, *J* 2.8 Hz, H-1^VIII^), 5.29 (s, 1H, H-1^VI^), 5.37 (s, 1H, H-1), 5.37 (s, 1H, H-1), 5.37–5.41 (m, 3H, H-2^VI^, 2 × H-1), 5.50 (d, 1H, *J* 4.4 Hz, H-3^V^), 5.58 (d, 1H, *J* 1.2 Hz, H-2^V^), 5.61–5.67 (m, 6H, H-3^II-IV^, H-2^II-IV^), 5.74–5.79 (m, 1H, H-3^I^), 5.77 (s, 1H, H-2^I^), 5.82 (s, 1H, H-1^I^), 6.80–6.86 (m, 2H, OC_6_H_4_O (H-3, H-5)), 7.02–7.08 (m, 2H, OC_6_H_4_O (H-2, H-6)), 7.19–7.27 (m, 8H, 4 × PhCO (H-3, H-5)), 7.34–7.55 (m, 24H, 7 × PhCO (H-3, H-5), 10 × PhCO (H-4)), 7.55–7.61 (m, 1H, PhCO (H-4)), 7.84–7.92 (m, 8H, 4 × PhCO (H-2, H-6)), 7.97–8.06 (m, 10H, 5 × PhCO (H-2, H-6)), 8.06–8.11 (m, 2H, PhCO (H-2, H-6)), 8.09–8.14 (m, 2H, PhCO (H-2, H-6)); ^13^C NMR (151 MHz, CDCl_3_): δ 11.97 (((CH_3_)_2_CH)_3_Si), 11.99 (((CH_3_)_2_CH)_3_Si), 12.01 (((CH_3_)_2_CH)_3_Si), 12.04 (((CH_3_)_2_CH)_3_Si), 12.2 (((CH_3_)_2_CH)_3_Si), 12.26 (2 × ((CH_3_)_2_CH)_3_Si), 12.28 (((CH_3_)_2_CH)_3_Si), 12.29 (((CH_3_)_2_CH)_3_Si), 12.4 (((CH_3_)_2_CH)_3_Si), 17.9 (((CH_3_)_2_CH)_3_Si), 17.96 (((CH_3_)_2_CH)_3_Si), 17.97 (((CH_3_)_2_CH)_3_Si), 18.02 (((CH_3_)_2_CH)_3_Si), 18.04 (((CH_3_)_2_CH)_3_Si), 18.06 (((CH_3_)_2_CH)_3_Si), 18.09 (((CH_3_)_2_CH)_3_Si), 18.13 (((CH_3_)_2_CH)_3_Si), 18.19 (((CH_3_)_2_CH)_3_Si), 18.20 (((CH_3_)_2_CH)_3_Si), 50.2 (CH_2_N_3_), 63.8 (C-5^VIII^, C-5^X^), 63.9 (C-5^VIII^, C-5^X^), 64.3 (C-5^IX^), 64.8 (C-5^VII^), 65.2 (C-5^V^, C-5^VI^), 65.3 (C-5^V^, C-5^VI^), 65.76 (2C, C-5^I^, C-5^II^, C-5^III^, C-5^IV^), 65.83 (C-5^I^, C-5^II^, C-5^III^, C-5^IV^), 65.9 (C-5^I^, C-5^II^, C-5^III^, C-5^IV^), 67.6 (CH_2_O), 77.10 (C-3^I^), 77.2 (2C, C-3^II^, C-3^III^, C-3^IV^, C-3^V^), 77.3 (C-3^II^, C-3^III^, C-3^IV^, C-3^V^), 77.6 (C-3^II^, C-3^III^, C-3^IV^, C-3^V^), 77.7 (C-3^IX^), 78.15 (C-2^VIII^ or C-2^X^), 78.24 (C-3^X^), 78.4 (C-2^VIII^ or C-2^X^), 78.51 (C-3^VII^, C-3^VIII^), 78.53 (C-3^VII^, C-3^VIII^), 81.0 (C-3^VI^), 81.5 (2C, C-2^II^, C-2^III^, C-2^IV^, C-2^V^), 81.59 (C-2^II^, C-2^III^, C-2^IV^, C-2^V^), 81.61 (C-2^II^, C-2^III^, C-2^IV^, C-2^V^), 81.9 (C-2^I^), 82.15 (C-4^II^, C-4^III^, C-4^IV^, C-4^V^, C-4^VI^), 82.17 (C-4^II^, C-4^III^, C-4^IV^, C-4^V^, C-4^VI^), 82.19 (C-4^II^, C-4^III^, C-4^IV^, C-4^V^, C-4^VI^), 82.26 (C-4^II^, C-4^III^, C-4^IV^, C-4^V^, C-4^VI^), 82.30 (C-4^II^, C-4^III^, C-4^IV^, C-4^V^, C-4^VI^), 82.8 (C-4^I^), 84.0 (C-2^VI^), 85.7 (C-4^VIII^, C-4^X^), 85.8 (C-4^VIII^, C-4^X^), 87.6 (C-4^VII^), 88.3 (C-4^IX^), 90.6 (C-2^VII^), 91.4 (C-2^IX^), 104.8 (C-1^VIII^), 104.9 (C-1^I^), 105.6 (C-1^X^), 105.8 (C-1^VI^), 105.87 (C-1^II^, C-1^III^, C-1^IV^, C-1^V^), 105.94 (C-1^II^, C-1^III^, C-1^IV^, C-1^V^), 105.97 (2C, C-1^II^, C-1^III^, C-1^IV^, C-1^V^), 106.2 (C-1^IX^), 106.9 (C-1^VII^), 115.6 (OC_6_H_4_O (C-3, C-5)), 118.3 (OC_6_H_4_O (C-2, C-6)), 128.10 (PhCO (C-3, C-5)), 128.14 (PhCO (C-3, C-5)), 128.16 (PhCO (C-3, C-5)), 128.20 (PhCO (C-3, C-5)), 128.3 (PhCO (C-3, C-5)), 128.4 (PhCO (C-3, C-5)), 128.46 (PhCO (C-3, C-5)), 128.49 (2 × PhCO (C-3, C-5)), 128.50 (PhCO (C-3, C-5)), 128.54 (PhCO (C-3, C-5)), 129.0 (PhCO (C-1)), 129.1 (PhCO (C-1)), 129.15 (PhCO (C-1)), 129.17 (3 × PhCO (C-1)), 129.22 (PhCO (C-1)), 129.28 (PhCO (C-1)), 129.34 (PhCO (C-1)), 129.4 (PhCO (C-1)), 129.76 (PhCO (C-2, C-6)), 129.78 (2 × PhCO (C-2, C-6)), 129.80 (PhCO (C-2, C-6)), 129.81 (PhCO (C-2, C-6)), 129.83 (3 × PhCO (C-2, C-6)), 129.86 (PhCO (C-2, C-6)), 129.92 (PhCO (C-2, C-6)), 130.0 (PhCO (C-2, C-6)), 132.7 (PhCO (C-4)), 132.8 (PhCO (C-4)), 132.9 (PhCO (C-4)), 133.0 (PhCO (C-4)), 133.1 (PhCO (C-4)), 133.2 (PhCO (C-4)), 133.2 (PhCO (C-4)), 133.3 (PhCO (C-4)), 133.4 (PhCO (C-4)), 133.44 (PhCO (C-4)), 133.49 (PhCO (C-4)), 150.7 (OC_6_H_4_O (C-1)), 153.6 (OC_6_H_4_O (C-4)), 165.01 (PhCO), 165.11 (PhCO), 165.14 (PhCO), 165.18 (PhCO), 165.23 (PhCO), 165.3 (PhCO), 165.37 (PhCO), 165.44 (PhCO), 165.5 (PhCO), 165.57 (PhCO), 165.64 (PhCO); ^29^Si INEPT NMR (60 MHz, CDCl_3_): δ 12.27, 12.83, 13.20, 13.44, 13.48, 13.50, 13.53, 13.76, 13.81, 13.85; HRMS (ESI): *m*/*z* [M + 2NH_4_]^2+^ Calcd for C_225_H_341_N_5_O_53_Si_10_^2+^ 2120.5912; Found: 2120.5897.


**4-(2-Azidoethoxy)phenyl 2,3,5-tri-*O*-acetyl-β**
**-d**
**-arabinofuranosyl-(1→**
**2)-3,5-di-*O*-acetyl-α**
**-d-**
**arabinofuranosyl-(1→**
**3)-[2,3,5-tri-*O*-acetyl-β**
**-d**
**-arabinofuranosyl-(1→**
**2)-3,5-di-*O*-acetyl-α**
**-d-**
**arabinofuranosyl-(1→**
**5)]-2-*O*-acetyl-α**
**-d-**
**arabinofuranosyl-(1→**
**5)-2,3-di-*O*-acetyl-α**
**-d-**
**arabinofuranosyl-(1→**
**5)-2,3-di-*O*-benzoyl-α-d-arabinofuranosyl-(1→**
**5)-2,3-di-*O*-acetyl-α-d-arabinofuranosyl-(1→**
**5)-2,3-di-*O*-acetyl-α-d-arabinofuranosyl-(1→**
**5)-2,3-di-*O*-acetyl-α-d-arabinofuranoside (21)**


Protected AEP decasaccharide **19** (25 mg, 0.059 mmol) was dissolved in THF (1 mL), then AcOH (4 μL, 0.059 mmol) and 1 M TBAF in THF (178 μL, 0.18 mmol) was added to the reaction mixture at 0 °C. The reaction mixture was stirred at 40 °C for 3 h. Then, the reaction mixture was concentrated under reduced pressure, co-evaporated with toluene (2 × 2 mL), dried in vacuo, and purified by silica gel column chromatography (gradient: MeOH in CH_2_Cl_2_, 2%→20%) to give benzoylated polyol of decasaccharide **20** (12 mg) isolated as a mixture with *n*-Bu_4_N^+^ salts according to MS and NMR data. *R*_f_ = 0.42 (CH_2_Cl_2_–MeOH 4:1); HRMS (ESI): *m*/*z* [M + Na]^+^ Calcd for C_135_H_133_N_3_NaO_53_ 2666.7696; found: 2666.7635. Then, the obtained reaction mixture was dissolved in MeOH (1 mL), followed by the addition of 1 M methanolic MeONa (50 µL, 0.05 mmol). The reaction mixture was kept at ~20 °C for 16 h. Then, the reaction mixture was neutralized with Dowex 50W×8 (H^+^) ion-exchange resin (the resin was washed with MeOH before addition) and then filtered. The resin was washed with MeOH (50 mL). The filtrate was concentrated under reduced pressure, and the residue was dried in vacuo. The obtained crude product was dissolved in anhydrous Py (1 mL), then Ac_2_O (1 mL) was added at 0 °C (ice–water bath). The reaction mixture was stirred at 20 °C for 24 h. The reaction was quenched by the addition of MeOH (1 μL) at 0 °C (ice–water bath), then concentrated under reduced pressure, co-evaporated with toluene (5 × 5 mL), and dried in vacuo. The residue was purified by silica gel chromatography (gradient: 5%→70% acetone in petroleum ether) to give acetylated decasaccharide **21** (2 mg, 14% over 3 steps). *R*_f_ = 0.20 (light petroleum–acetone 1:1); ${\left[\alpha \right]}_{D}^{23}$ +53.9 (c 0.15 in CHCl_3_); ^1^H NMR (600 MHz, CDCl_3_): δ 2.08 (s, 9H, 3 × CH_3_CO), 2.08 (s, 9H, 3 × CH_3_CO), 2.09 (s, 6H, 2 × CH_3_CO), 2.09 (s, 3H, CH_3_CO), 2.10 (s, 3H, CH_3_CO), 2.11 (s, 3H, CH_3_CO), 2.11 (s, 15H, 5 × CH_3_CO), 2.11 (s, 3H, CH_3_CO), 2.12 (s, 6H, 2 × CH_3_CO), 2.14 (s, 3H, CH_3_CO), 2.14 (s, 3H, CH_3_CO), 3.58 (t, 2H, *J* 5.0 Hz, CH_2_N_3_), 3.70–3.78 (m, 6H, H-5^I-VI^a), 3.86–3.90 (m, 1H, H-5^VI^b), 3.92–3.98 (m, 5H, H-5^I-V^b), 4.11–4.13 (m, 3H, CH_2_O), 4.10–4.26 (m, 15H, H-4^II-X^, H-3^VI^, H-5^VII-X^a, H-5^VII^b or H-5^IX^b), 4.25 (d, 1H, *J* 2.0 Hz, H-2^VII^), 4.29 (d, 1H, *J* 1.7 Hz, H-2^IX^), 4.32 (dd, 1H, *J* 11.5 Hz, *J* 3.9 Hz, H-5^VII^b or H-5^IX^b), 4.34–4.39 (m, 3H, H-4^I^, H-5^VIII^b, H-5^X^b), 4.95–5.01 (m, 4H, H-2^VIII^, H-2^X^, H-3^VII^, H-3^IX^), 5.02 (s, 1H, H-1^VII^), 5.04–5.07 (m, 3H, H-3^V^, H-2^VI^, H-1^IX^), 5.12–5.18 (m, 12H, H-1^II-VI^, H-2^II-V^, H-3^II-IV^), 5.29 (dd, 1H, *J* 5.3 Hz, *J* 1.9 Hz, H-3^I^), 5.32–5.37 (m, 3H, H-3^VIII^, H-3^X^, H-2^I^), 5.39 (d, 1H, *J* 4.7 Hz, H-1^X^), 5.44 (d, 1H, *J* 4.7 Hz, H-1^VIII^), 5.61 (s, 1H, H-1^I^), 6.84–6.88 (m, 2H, OC_6_H_4_O (H-3, H-5)), 6.98–7.03 (m, 2H, OC_6_H_4_O (H-2, H-6)); ^13^C NMR (151 MHz, CDCl_3_): δ 20.3 (CH_3_CO), 20.4 (CH_3_CO), 20.6 (CH_3_CO), 20.67 (CH_3_CO), 20.69 (CH_3_CO), 20.73 (CH_3_CO), 20.75 (CH_3_CO), 20.80 (CH_3_CO), 50.2 (CH_2_N_3_), 63.4 (C-5^VII^, C-5^IX^), 63.7 (C-5^VII^, C-5^IX^), 64.2 (C-5^VI^), 65.0 (C-5^I-V^, C-5^VIII^, C-5^X^), 65.1 (C-5^I-V^, C-5^VIII^, C-5^X^), 65.18 (C-5^I-V^, C-5^VIII^, C-5^X^), 65.24 (3C, C-5^I-V^, C-5^VIII^, C-5^X^), 65.7 (C-5^I-V^, C-5^VIII^, C-5^X^), 67.6 (CH_2_O), 75.5 (C-3^VIII^, C-3^X^), 75.7 (C-3^VIII^, C-3^X^), 76.4 (C-3^I^), 76.6 (C-3^II-V^, C-3^VII^, C-3^IX^, C-2^VIII^, C-2^X^), 76.7 (C-3^II-V^, C-3^VII^, C-3^IX^, C-2^VIII^, C-2^X^), 76.8 (C-3^II-V^, C-3^VII^, C-3^IX^, C-2^VIII^, C-2^X^), 76.9 (C-3^II-V^, C-3^VII^, C-3^IX^, C-2^VIII^, C-2^X^), 77.1 (C-3^II-V^, C-3^VII^, C-3^IX^, C-2^VIII^, C-2^X^), 77.4 (C-3^II-V^, C-3^VII^, C-3^IX^, C-2^VIII^, C-2^X^), 77.7 (C-3^II-V^, C-3^VII^, C-3^IX^, C-2^VIII^, C-2^X^), 78.9 (C-4^VIII^, C-4^X^), 78.6 (C-4^VIII^, C-4^X^), 80.3 (C-4^II-VII^, C-4^IX^, C-3^VI^, C-2^II-V^), 80.7 (C-4^II-VII^, C-4^IX^, C-3^VI^, C-2^II-V^), 80.8 (C-4^II-VII^, C-4^IX^, C-3^VI^, C-2^II-V^), 81.1 (C-4^II-VII^, C-4^IX^, C-3^VI^, C-2^II-V^), 81.2 (C-4^II-VII^, C-4^IX^, C-3^VI^, C-2^II-V^), 81.47 (C-4^II-VII^, C-4^IX^, C-3^VI^, C-2^II-V^), 81.51 (C-4^II-VII^, C-4^IX^, C-3^VI^, C-2^II-V^), 81.53 (C-4^II-VII^, C-4^IX^, C-3^VI^, C-2^II-V^), 81.68 (2C, C-4^II-VII^, C-4^IX^, C-3^VI^, C-2^II-V^), 81.73 (C-4^II-VII^, C-4^IX^, C-3^VI^, C-2^II-V^), 81.9 (C-4^I^), 82.0 (C-2^I^), 82.2 (C-4^II-VII^, C-4^IX^, C-3^VI^, C-2^II-V^), 83.4 (C-2^VI^), 83.9 (C-2^IX^), 84.2 (C-2^VII^), 99.3 (C-1^X^), 99.5 (C-1^VIII^), 104.8 (C-1^I^), 105.2 (C-1^II-VII^, C-1^IX^), 105.35 (3C, C-1^II-VII^, C-1^IX^), 105.39 (C-1^II-VII^, C-1^IX^), 105.43 (C-1^II-VII^, C-1^IX^), 105.7 (C-1^II-VII^, C-1^IX^), 115.6 (OC_6_H_4_O (C-3, C-5)), 118.3 (OC_6_H_4_O (C-2, C-6)), 150.4 (OC_6_H_4_O (C-1)), 153.8 (OC_6_H_4_O (C-4)), 169.7 (CO), 169.7 (CO), 169.9 (CO), 170.00 (CO), 170.02 (CO), 170.1 (CO), 170.2 (CO), 170.18 (CO), 170.20 (CO), 170.3 (CO), 170.46 (CO), 170.51 (CO), 170.6 (CO); HRMS (ESI): *m*/*z* [M + 2NH_4_]^2+^ Calcd for C_100_H_139_N_5_O_63_^2+^ 1208.8908; found: 1208.8948.

## 5. Conclusions

In summary, a synthesis of the branched α-(1→3)-, α-(1→5), β-(1→2)-linked hexaarabinofuranoside with CEP aglycone using silylated Ara-β-(1→2)-Ara disaccharide was accomplished. The CEP aglycone in hexaarabinofuranoside was cleaved, and the obtained hemiacetal was converted to the new glycosyl donor with *N*-phenyltrifluoroacetimidoyl leaving group. The coupling of *N*-phenyltrifluoroacetimidate of hexaarabinofuranoside and benzoylated tetraarabinofuranoside α-(1→5)-linked glycosyl acceptor promoted by TfOH successfully led to the branched α-(1→5)-, α-(1→3)-, β-(1→2)-linked decaarabinofuranoside with CEP aglycone. After the replacement of the chlorine atom with an azido group in the aglycone of the resulting decaarabinofuranoside and following removal of silyl and acyl groups, the deprotected decaarabinofuranoside bearing 4-(2-azidoethoxy)phenyl aglycone (AEP) was obtained.

The synthesized decaarabinofuranoside and neoglycoconjugates derived from them expand the library of antigens based on oligoarabinofuranosides related to LAM and AG fragments of mycobacteria developed by us. It is important to note that such libraries hold promise for the creation of new multi-antigen/multi-epitope assays for the serodiagnosis of tuberculosis and leprosy, as it has been established that tests based on a single antigen may not be equally sensitive in all regions endemic for tuberculosis and leprosy [10].

## Data Availability

The original contributions presented in this study are included in the article/Appendix A. Further inquiries can be directed to the corresponding author.

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
