# Peer review of "A Rational Synthesis of a Branched Decaarabinofuranoside Related to the Fragments of Mycobacterial Polysaccharides"

_molecules, 2025, doi:10.3390/molecules30153295_

Round 1

Reviewer 1 Report

Comments and Suggestions for Authors

This manuscript describes “A Rational Synthesis of a Branched Decaarabinofuranoside Related to the Fragments of Mycobacterial Polysaccharides “. It seems to be simple extension of the authors' previous method to synthesize hexasaccharide to decasaccharide. However, there has been still some difficulties to apply to the larger oligosaccharide synthesis as shown in this manuscript about the trial and achievement. Detailed experimental support shown fully might help to understand the results. It is well written and should be published in Molecules, however, following issues should be considered before publication.

Comments:

[1] Page 6 Scheme 3: Did the author try perTIPS ether of thioglycoside of heptasaccharide as donor for the fragment coupling which could be converted from 2? If not, why did not use the same type TIPS-protected donor for further elongation?

[2] Page 7, 2.2.1: The first paragraph is written in different font and size.

[3] Page 7, 2.2.1: Compound numbers should be written in Bold.

[4] Page 8, line 2 and 18: Compound numbers should be written in Bold.

[5] Page 8, line 4 and 12: 1H should be written as 1H.

[6] Page 8, line 10: How did you calculate the yield of this reaction? Did you start from the mixture of 15 and 16 (1 : 0.4)? Was the configuration of 16 β-anomer? If α-anomer of 16 has been formed, is it possible to give 1-O-imidate from α-16?

[7] Do you need to put numbers for arabinose in the figures? If so add all (Scheme 1, Scheme 2,  8 in scheme 3, 20 in scheme 6).

[8] Page 8, 2.2.2: Why do not the authors put the figure or scheme from 15 and 16 to 19?  

[9] Page 8, 2.2.2, line 2–3: Arrows seem to be overlapped somehow.  

[10] Scheme 6. From 11 to 5: The yield of global deprotection is very low. All steps can be expected to give full conversion. The authors should comment which deprotection step did not give good conversion yield.   

[11] Although acetate is suitable to assign the configuration of the synthetic decasaccharide easily, the author should discuss the NMR confirmation of 5, which would be more interesting than its acetate to the reader.

[12] Page 10, line 26: In the discussion part, the authors should discuss why TfOH instead of AfgOTf was effective for (1→3)-α-arabinoburanosylation and fragment coupling as well as polymerization.

[13] Page 10, line 39: Would you also explain the length and size of oligo-arabinofuranosides for effective antigen to conjugate for tuberculosis screening.

[14] “pyridine” was used in page 4. But not in scheme caption and experimental section. Py seems not to be general abbreviation for pyridine. In the first use, “pyridine (Py)” should be written.

[15] Both oligo-arabinofuranoside and oligoarabinofuranoside were found which should be written in the same way.

[16] “J” should be written in italic for coupling constants. Check the author’s info.

[17] Page 14, compound 6: “p”- should be written in italic.

[18] Page 15, line 5: “−5°С” should be “−5 °С”.

[19] Page 15, line 16 :“9[33]” should be “9 [33]”. Add a space. Add a space similarly before indication of ref number or any other words in all cases except “,” , “.” And so on, found many in the manuscript.

[20] Page 15, line 16: [33] for indication of ref should not be written in Bold here.

[21] Page 15, line 30: CH3COO should be CH3COO

[22] Page 16, line 2: Add space after period.

[23] Page 18, for mixture of compound 15 and 16: It is difficult to understand what are 6.60 (d, 1H, J 4.5 Hz, H-1I minor), 97.04 (C- 1I minor). Are they the peaks for 16? If 16, write 16 instead of minor. Here the authors should also write the ratio (15:16).

[24] Integration of TIPS in 1H NMR is shown as 614 H which means 29.23 of TIPS groups. Is it theoretically correct?

[25] "gradient: petroleum ether–EtOAc, 3%→8%" and "gradient: 0%→30% EtOAc in petroleum ether" are found, which should be written in the same way.

[26] Page 23, conclusion line 3: “the terminal fragments” should be “the non-reducing terminal fragment”.

[27] Page 23, conclusion: Synthesis of hexasaccharide of CEP is not shown in the manuscript. In the abstract, there is a comment about synthesis of hexasaccharide of CEP as well.

[28] Sup. Data: there are two contents. Remove one.

[29] P2: compound 4: “tris” should be “Tris”

[30] P23: compound 15: “tris” should be “Tris”

[30] 2D Spectra for H-H COSY and HSQC should be displayed at higher resolution.

Author Response

Reviewer #1:

This manuscript describes “A Rational Synthesis of a Branched Decaarabinofuranoside Related to the Fragments of Mycobacterial Polysaccharides “. It seems to be simple extension of the authors' previous method to synthesize hexasaccharide to decasaccharide. However, there has been still some difficulties to apply to the larger oligosaccharide synthesis as shown in this manuscript about the trial and achievement. Detailed experimental support shown fully might help to understand the results. It is well written and should be published in Molecules, however, following issues should be considered before publication.

 REPLY:

We thank the Reviewer #1 for high evaluation of our manuscript.

Comments:

[1] Page 6 Scheme 3: Did the author try perTIPS ether of thioglycoside of heptasaccharide as donor for the fragment coupling which could be converted from 2? If not, why did not use the same type TIPS-protected donor for further elongation?

REPLY:

We would like to thank the Reviewer #1 for the question, and we have taken into consideration the possibility of the conversion of hexasaccharide 2 to perTIPS thioglycoside. Such a glycosyl donor can be expected to be effective in a-arabinofuranosylation. However, the actual stereoselectivity is still unknown. On the other hand, it should be noted that the use of glycosyl donor 4 generated from 2 and having a 2-O-acyl participating group ensures the creation of 1,2-trans-(a) glycosidic linkage without extra deprotecting/protecting steps.

[2] Page 7, 2.2.1: The first paragraph is written in different font and size.

REPLY:

The manuscript was modified

[3] Page 7, 2.2.1: Compound numbers should be written in Bold.

The manuscript was modified accordingly.

REPLY:

[4] Page 8, line 2 and 18: Compound numbers should be written in Bold.

REPLY:

The manuscript was modified accordingly.

[5] Page 8, line 4 and 12: 1H should be written as 1H.

REPLY:

The manuscript was modified.

[6] Page 8, line 10: How did you calculate the yield of this reaction? Did you start from the mixture of 15 and 16 (1 : 0.4)? Was the configuration of 16 β-anomer? If α-anomer of 16 has been formed, is it possible to give 1-O-imidate from α-16?

REPLY:

We revised the ratio of isomers 15 and16. Hemiacetal 15 undergoes mutarotation in the solution. The initial ratio of 15a : 15b : 16 was 1 : 0.45 : 0.28 according to NMR data. The yield of this reaction was calculated for the sum of a-15, b-15 and the product of benzoate migration (16), which are all isomeric hence having equal MW. The large J1,2 = 4.5 Hz indicates that 16 has b configuration. No migration product with a configuration was detected. Apparently, 16 undergoes reverse migration of the anomeric benzoate to give 15 in the course of imidate synthesis. The manuscript was modified.

[7] Do you need to put numbers for arabinose in the figures? If so add all (Scheme 1, Scheme 2,  8 in scheme 3, 20 in scheme 6).

REPLY:

The Schemes 1, 2, 3 were replaced and the numbers for arabinofuranose residues were added.

[8] Page 8, 2.2.2: Why do not the authors put the figure or scheme from 15 and 16 to 19?  

REPLY:

Scheme 5 with compounds 15, 16 and 19 was inserted into the text.

[9] Page 8, 2.2.2, line 2–3: Arrows seem to be overlapped somehow.

REPLY: 

The text was modified.

[10] Scheme 6. From 11 to 5: The yield of global deprotection is very low. All steps can be expected to give full conversion. The authors should comment which deprotection step did not give good conversion yield.   

REPLY:

Deprotection of complex oligosaccharides is a challenging task. It should be noted that 10 triisopropylsilyl groups present in decaarabinofuranoside are to be removed simultaneously. We anticipate that significant losses may occur during the removal of TIPS groups and following purification from excess reagent and side products. In our future investigation, we intend to optimize this step. The text was modified.

[11] Although acetate is suitable to assign the configuration of the synthetic decasaccharide easily, the author should discuss the NMR confirmation of 5, which would be more interesting than its acetate to the reader.

REPLY:

We added an additional NMR description for deprotected decaarabinofuranoside.

[12] Page 10, line 26: In the discussion part, the authors should discuss why TfOH instead of AgOTf was effective for (1→3)-α-arabinoburanosylation and fragment coupling as well as polymerization.

REPLY:

NIS/AgOTf as a mild promoter system and the use of it led to the product of glycosylation only of the more reactive primary position of glycosyl acceptor. NIS/TfOH is a stronger promoter system and favors the glycosylation at both the primary position and the less reactive secondary position of glycosyl acceptor. It should be noted that the nature of protective groups in a glycosyl donor can also significantly affect the outcome of glycosylation. Due to higher reactivity of the silylated disaccharide Ara-b-(1→2)-Ara glycosyl donors, such as 6, side reactions are suppressed and we didn’t observe the formation of the oligomerization products.

[13] Page 10, line 39: Would you also explain the length and size of oligo-arabinofuranosides for effective antigen to conjugate for tuberculosis screening.

REPLY:

We believe that the length of oligoarabinofuranosides is important for effective tuberculosis screening. The synthesized decaarabinofuranoside and neoglycoconjugates derived from them expand our library of antigens based on oligoarabinofuranosides related to LAM and AG fragments of mycobacteria. It is important to note that such libraries hold promise for the creation of new multi-antigen/multi-epitope assays for the serodiagnosis of tuberculosis and leprosy, as it has been established that tests based on a single antigen may not be equally sensitive in all regions endemic for tuberculosis and leprosy.

The text was modified.

 [14] “pyridine” was used in page 4. But not in scheme caption and experimental section. Py seems not to be general abbreviation for pyridine. In the first use, “pyridine (Py)” should be written.

REPLY:

The text was modified.

[15] Both oligo-arabinofuranoside and oligoarabinofuranoside were found which should be written in the same way.

REPLY:

We believe that "oligoarabinofuranoside" is correct. The text was modified.

[16] “J” should be written in italic for coupling constants. Check the author’s info.

REPLY:

The text was modified.

[17] Page 14, compound 6: “p”- should be written in italic.

REPLY:

The text was modified.

[18] Page 15, line 5: “−5°С” should be “−5 °С”.

REPLY:

The text was modified.

[19] Page 15, line 16 :“9[33]” should be “9 [33]”. Add a space. Add a space similarly before indication of ref number or any other words in all cases except “,” , “.” And so on, found many in the manuscript.

REPLY:

The text was modified.

[20] Page 15, line 16: [33] for indication of ref should not be written in Bold here.

REPLY:

The text was modified.

[21] Page 15, line 30: CH3COO should be CH3COO

REPLY:

We replaced CH3COO with CH3COO.

[22] Page 16, line 2: Add space after period.

REPLY:

The space was added.

[23] Page 18, for mixture of compound 15 and 16: It is difficult to understand what are 6.60 (d, 1H, J 4.5 Hz, H-1I minor), 97.04 (C- 1I minor). Are they the peaks for 16? If 16, write 16 instead of minor. Here the authors should also write the ratio (15:16).

REPLY:

We agree with Reviewer #1. The text was modified as following: The ratio of 15a : 15b : 16 = 1 : 0.45 : 0.28 according to NMR data and was determined by integration of signals of corresponding anomeric protons of the reducing residues (I) in the 1H NMR spectrum (5.62 (s, 1H, H-1Iα) for 15a, 5.72 (t, 1H, J 5.1 Hz, H-1Iβ) for 15b and 6.60 (d, 1H, J 4.5 Hz, H-1I) for 16. The formation of the b-linked product 16 of the migration of benzoyl group follows from low-field position of the anomeric proton: dH 6.60 (d, 1H, J 4.5 Hz, H-1I) in 1H NMR spectrum, which correlated with the signal of anomeric carbon atom of monosaccharide residue dC 97.04 (C-1I). The text was modified.

[24] Integration of TIPS in 1H NMR is shown as 614 H which means 29.23 of TIPS groups. Is it theoretically correct?

REPLY:

We checked the ratio of 15:16 and provided the correct NMR data.

[25] "gradient: petroleum ether–EtOAc, 3%→8%" and "gradient: 0%→30% EtOAc in petroleum ether" are found, which should be written in the same way.

REPLY:

The "gradient: 0%→30% EtOAc in petroleum ether" was changed to: "gradient: petroleum ether, 0%→30% ".

[26] Page 23, conclusion line 3: “the terminal fragments” should be “the non-reducing terminal fragment”.

REPLY:

The conclusion was modified. See [27].

[27] Page 23, conclusion: Synthesis of hexasaccharide of CEP is not shown in the manuscript. In the abstract, there is a comment about synthesis of hexasaccharide of CEP as well.

REPLY:

The abstract was modified: A rational synthesis of the branched decaarabinofuranoside with 4-(2-azidoethoxy)phenyl aglycone (a Janus aglycone) related to the non-reducing terminal fragments of the arabinogalactan and lipoarabinomannan from Mycobacterium tuberculosis was proposed.

The conclusion was modified: In summary, a synthesis of the branched a-(1→3)-, a-(1→5), b-(1→2)-linked hexaarabinofuranoside with 4-(2-chloroethoxy phenyl (CEP) aglycone, its conversion to the new glycosyl donor and to the branched decaarabinofuranoside with 4-(2-azidoethoxy)phenyl (AEP) aglycone was accomplished. The obtained decaarabinofuranoside related to the the non-reducing terminal fragment of the arabinogalactan and lipoarabinomannan from Mycobacterium tuberculosis can be used for further preparation of conjugates as antigens, which are important for creating tuberculosis screening assays.

 [28] Sup. Data: there are two contents. Remove one.

REPLY:

We thank the Reviewer #1. We attached only one content.

[29] P2: compound 4: “tris” should be “Tris”

REPLY:

We replaced “tris” with “Tris”.

[30] P23: compound 15: “tris” should be “Tris”

We replaced “tris” with “Tris”.

REPLY:

[30] 2D Spectra for H-H COSY and HSQC should be displayed at higher resolution.

2D spectra for H-H COSY and HSQC with the best resolution available for us.

Reviewer 2 Report

Comments and Suggestions for Authors

Referee’s opinion for the following Manuscript:

A Rational Synthesis of a Branched Decaarabinofuranoside Related to the Fragments of Mycobacterial Polysaccharides

Polina I. Abronina,*[1] Nelly N. Malysheva,[1] Maxim Y. Karpenko,[1] Dmitry S. Novikov,[1] Alexander I. Zinin,[1] N. G. Kolotyrkina,[a] Leonid O. Kononov,*[1]

The manuscript describes the synthesis of branched hexa- and decaarabinofuranosides with 4-(2-azidoethoxy)phenyl (AEP) aglycone related to the terminal fragments of the arabinogalactan and lipoarabinomannan from Mycobacterium tuberculosis. The manuscript follows the formal requirements of Molecules, but is hard to follow. This is partly because some English revision is needed. Several questions have emerged while reviewing the manuscript, as listed below, that all in all require a major revision.

The reviewer also wants to highlight for the authors that the term “Janus aglycone” which (if the reviewer is right) the authors started using as a term, describes the structural element elegantly.

Minor remarks:

The first 3 citations should be corrected, so that citation numbers in square brackets in the manuscript are before punctuations and not in superscript (e.g. “[1],”).

-The Introduction is lengthy (3 pages are long) and though many synthetic approaches are described, it lacks a Scheme showing, at least in general, the described modifications. The Figure that is in the Introduction is miss-titled as it describes the use of the Janus aglycone, but it shows the products. Overall, the reviewer recommends revising the Introduction to make it more concise and include a scheme/figure that illustrates the previously described methods by others, highlight the research group's previous work, and outline the intended plan.

-Bu4N+ should be corrected in the manuscript to n-Bu4N+ (if the n-Bu version was used, or use the appropriate abbreviation).

-In case of the synthesis of compound 2, is the first, two-step method in any way better than the one-step version utilising TfOH/NIS promotion? If not, the superiority of the one-step method should be mentioned. If there is a reason why it is not mentioned, please elaborate.

-Why are the authors highlighting that only the α-configuration of the glycosidic linkage can be observed when synthesizing compound 2?

-The style of the first paragraph under 2.2.1 needs formatting.

-When describing the 1:1 α:β anomer ratio in the case of 15, please also mention/highlight the ppm of the group(s) in question.

-Under Scheme 6, the description of reactions seems off or has an error with step c) being after the yield over 3 steps.

Major remarks:

-When describing that no products of the cleavage of the inter-saccharide glycosidic bond were observed during thiolysis of 10, the authors also mention 11, but no thiolysis of 11 is described in the manuscript. Is it missing?

-Scheme 5 is missing!

-The authors highlight the optimization of the synthesis of compound 6 compared to the previously reported. This is somewhat true. For the basis of the comparison, the authors choose literature [1] where the reported method’s overall yield is 25% (taking into consideration of previous method described in [2]). However, regarding the synthesis starting from 11 based on their previous methods [1,2], an overall yield of 43% for 11 can be calculated, resulting in a final yield of 11%. For 9 based also on previously reported methods [1,2], a yield of 52% can be calculated, giving a final yield of 27% meaning only minimal yield optimization and one more step. It is also overshadowed by the fact that in [2], no yield for several compounds is reported, leaving a gap for even more reduction of the final yield. If provided enough published results, the “significant optimization” can be used; otherwise, “an alternative method” should be used.

-Regarding the previous comment, highlighting that the synthesis of 6 required more steps based on [1] is not applicable.

-The part of oligomerization is hard to understand without Scheme 5.

-Regarding Materials and Methods NMRs of compounds starting from materials with TIPS protecting groups (2,4,10,13,14,16,18,19) have in some cases high, some cases less amount of triisopropylsilane contamination around 1 ppm, overshadowing the yields. Please provide purified NMRs, or yields cannot be accepted.

[1] 10.1002/ejoc.202201110

[2] 10.1016/j.carres.2014.05.017

If addressed properly, the work presented could be accepted for publication.

Author Response

Reviewer #2: 

The manuscript describes the synthesis of branched hexa- and decaarabinofuranosides with 4-(2-azidoethoxy)phenyl (AEP) aglycone related to the terminal fragments of the arabinogalactan and lipoarabinomannan from Mycobacterium tuberculosis. The manuscript follows the formal requirements of Molecules, but is hard to follow. This is partly because some English revision is needed. Several questions have emerged while reviewing the manuscript, as listed below, that all in all require a major revision.

The reviewer also wants to highlight for the authors that the term “Janus aglycone” which (if the reviewer is right) the authors started using as a term, describes the structural element elegantly.

REPLY:

We propose to call 4-(w-chloroalkoxy)phenyl aglycones due to dual function the Janus aglycone (and corresponding glycosides – Janus glycosides) in analogy to the well-known Janus particles (Walther, A.; Muller, A.H. Janus particles: synthesis, self-assembly, physical properties, and applications. Chem. Rev. 2013, 113, 5194-5261; https://doi.org/10.1021/cr300089t). Janus particles are a special types of nanoparticles whose surfaces have two or more distinct physical properties hence two different types of chemistry may occur on the same particle.

Minor remarks:

The first 3 citations should be corrected, so that citation numbers in square brackets in the manuscript are before punctuations and not in superscript (e.g. “[1],”).

REPLY:

The text was modified.

-The Introduction is lengthy (3 pages are long) and though many synthetic approaches are described, it lacks a Scheme showing, at least in general, the described modifications. The Figure that is in the Introduction is miss-titled as it describes the use of the Janus aglycone, but it shows the products. Overall, the reviewer recommends revising the Introduction to make it more concise and include a scheme/figure that illustrates the previously described methods by others, highlight the research group's previous work, and outline the intended plan.

REPLY:

We were trying to improve the introduction to our research. Since most of the existing methods focus on the use of benzyl-containing glycosyl donors for 1,2-cis-arabinofuranosylation, we aimed on the demonstrating the benzyl-free strategy developed by us. This strategy is based on the use of silylated arabinofuranosyl donors (we inserted an additional figure). The use of such glycosyl donors facilitates the synthesis of non-reducing terminal oligoarabinoside fragments of mycobacterial polysaccharides with an azido group in the aglycone.

The introduction was modified.

Bu4N+ should be corrected in the manuscript to n-Bu4N+ (if the n-Bu version was used, or use the appropriate abbreviation).

REPLY:

The Bu4N+  was replaced with n-Bu4N+.

-In case of the synthesis of compound 2, is the first, two-step method in any way better than the one-step version utilising TfOH/NIS promotion? If not, the superiority of the one-step method should be mentioned. If there is a reason why it is not mentioned, please elaborate.

REPLY:

We can conclude that the use of (NIS/TfOH) is more preferable for bis-glycosylation.

-Why are the authors highlighting that only the α-configuration of the glycosidic linkage can be observed when synthesizing compound 2?

REPLY:

Stereoselective formation of 1,2-trans (a)-arabinofuranosidic linkage can be easy achieved by glycosylation with monosaccharide glycosyl donor containing a participating acyl group at O-2. Diarabinofuranoside could be considered as a monosaccharide glycosyl donor containing a bulky carbohydrate substituent at O-2 instead of participating acyl group at O-2, generally resulting in the loss of stereocontrol. Indeed, in the case of the use of Ara-b-(1→2)-Ara disaccharide glycosyl donor, which contains only O-benzoyl substituents, the absence of stereocontrol (a:b = 1:2) was observed (10.1002/ejoc.202201110). On the contrary, the use of Ara-b-(1→2)-Ara tolyl thioglycoside containing five TIPS groups was effective in stereoselective 1,2-trans (a) arabinofuranosylation.

-The style of the first paragraph under 2.2.1 needs formatting.

REPLY:

We formatted the first paragraph of 2.2.1.  

-When describing the 1:1 α:β anomer ratio in the case of 15, please also mention/highlight the ppm of the group(s) in question.

REPLY:

We checked the ratio of 15:16 and provided the correct NMR data. The ratio of 15a : 15b : 16 = 1 : 0.45 : 0.28 according to NMR data, which was determined by integration of signals of corresponding anomeric protons of the residues I in 1H NMR spectrum (5.62 (s, 1H, H-1Iα) for 15a, 5.72 (t, 1H, J 5.1 Hz, H-1Iβ) for 15b  and 6.60 (d, 1H, J 4.5 Hz, H-1I) for 16.

-Under Scheme 6, the description of reactions seems off or has an error with step c) being after the yield over 3 steps.

 REPLY:

The description for Scheme 6 was changed.

Major remarks:

-When describing that no products of the cleavage of the inter-saccharide glycosidic bond were observed during thiolysis of 10, the authors also mention 11, but no thiolysis of 11 is described in the manuscript. Is it missing?

 REPLY:

The text was modified as following: To the best of our knowledge, the products of the cleavage of the inter-saccharide glycosidic bond were not observed during thiolysis of 10 and 13.

-Scheme 5 is missing!

REPLY:

We are very grateful to the reviewer. Scheme 5 was inserted to the text.

-The authors highlight the optimization of the synthesis of compound 6 compared to the previously reported. This is somewhat true. For the basis of the comparison, the authors choose literature [1] where the reported method’s overall yield is 25% (taking into consideration of previous method described in [2]). However, regarding the synthesis starting from 11 based on their previous methods [1,2], an overall yield of 43% for 11 can be calculated, resulting in a final yield of 11%. For 9 based also on previously reported methods [1,2], a yield of 52% can be calculated, giving a final yield of 27% meaning only minimal yield optimization and one more step. It is also overshadowed by the fact that in [2], no yield for several compounds is reported, leaving a gap for even more reduction of the final yield. If provided enough published results, the “significant optimization” can be used; otherwise, “an alternative method” should be used.

REPLY:

We agree with reviewer that an alternative synthesis of diarabinofuranoside 6 was proposed.

-Regarding the previous comment, highlighting that the synthesis of 6 required more steps based on [1] is not applicable.

REPLY:

The text was modified.

-The part of oligomerization is hard to understand without Scheme 5.

REPLY:

We are very grateful to the Reviewer #2. The Scheme 5 was inserted to the text.

-Regarding Materials and Methods NMRs of compounds starting from materials with TIPS protecting groups (2,4,10,13,14,16,18,19) have in some cases high, some cases less amount of triisopropylsilane contamination around 1 ppm, overshadowing the yields. Please provide purified NMRs, or yields cannot be accepted.

 REPLY:

We are well aware of the problem, especially because the side product of silylation (triisopropylsilanol) is difficult to detect by TLC. We carefully controlled the presence of triisopropylsilanol in our silylation products (steps 910 and 12 13) using 29Si NMR, and re-chromatographed the contaminated fractions if needed. It should be noted that only the signals of TIPS ether groups are found in the 29Si NMR spectra we presented. The signals of triisopropylsilanol at 15.0 ppm in the 29Si NMR spectra for all obtained compounds (including 10 and 13) are absent. It is important that NMR data for 6 (10.1002/ejoc.202201110) are in accordance with the expected data. Besides, at the next step the TIPS groups remain unaffected. The 1H NMR spectrum for 15+16 was replaced and the ratio of 15+16 was revised. We believe that the given values are within the expected range of integration error. Specifically, we propose that integration errors may occur for protons with different relaxation times.

 If addressed properly, the work presented could be accepted for publication.

 REPLY:

We would like to thank the Reviewer #2 for the opportunity to revise our manuscript.

Round 2

Reviewer 2 Report

Comments and Suggestions for Authors

All my questions have been adequately answered.

Author Response

The following minor revisions are requested from the authors: i) ¹³C NMR Chemical Shifts: Please report the chemical shifts for ¹³C NMR using only one decimal place (e.g., 128.4 ppm), unless a second decimal digit is needed to distinguish between closely spaced signals (e.g., 128.41 and 128.47 ppm).

REPLY:

The ¹³C NMR notation was changed.

  1. ii) High-Resolution Mass Spectrometry (HRMS): The HRMS data for the ion [M + 2NH₄]²⁺ appear to be inconsistent. The molecular formula is currently given as C₂₂₅H₃₄₁N₅O₅₃Si₁₀⁺, which suggests a monocation, while the ion is clearly a dication. To accurately reflect the observed species, the formula should be corrected to C₂₂₅H₃₄₁N₅O₅₃Si₁₀²⁺. The authors must revise the charge notation and the corresponding m/z values accordingly. Please also review the manuscript for any similar inconsistencies and make the necessary corrections.

REPLY:

We agree with Reviewer. We revised the charge notation and the corresponding m/z values.

 iii) Conclusions: Considering the scope of the study, the conclusions in their current form appear rather limited. The authors are encouraged to expand and strengthen the concluding remarks to better highlight the significance and impact of the work. iv) Reviewer 2's

Comments: The authors must ensure that pertinebt comments and suggestions provided by Reviewer 2 have been carefully considered and adequately addressed in the revised version.

REPLY:

The Conclusion was changed.